# Graph Coarsening with Message-Passing Guarantees

**Antonin Joly**
IRISA, Rennes, France
`antonin.joly@inria.fr`

**Nicolas Keriven**
CNRS, IRISA, Rennes, France
`nicolas.keriven@cnrs.fr`

## Abstract

Graph coarsening aims to reduce the size of a large graph while preserving some of its key properties, which has been used in many applications to reduce computational load and memory footprint. For instance, in graph machine learning, training Graph Neural Networks (GNNs) on coarsened graphs leads to drastic savings in time and memory. However, GNNs rely on the Message-Passing (MP) paradigm, and classical spectral preservation guarantees for graph coarsening do not directly lead to theoretical guarantees when performing naive message-passing on the coarsened graph.

In this work, we propose a new message-passing operation specific to coarsened graphs, which exhibit theoretical guarantees on the preservation of the propagated signal. Interestingly, and in a sharp departure from previous proposals, this operation on coarsened graphs is often oriented, even when the original graph is undirected. We conduct node classification tasks on synthetic and real data and observe improved results compared to performing naive message-passing on the coarsened graph.

## 1   Introduction

In recent years, several applications in data science and machine learning have produced large-scale *graph* data [20, 5]. For instance, online social networks [13] or recommender systems [40] routinely produce graphs with millions of nodes or more. To handle such massive graphs, researchers have developed general-purpose *graph reduction* methods [4], such as **graph coarsening** [32, 7]. It consists in producing a small graph from a large graph while retaining some of its key properties, and starts to play an increasingly prominent role in machine learning applications [7].

**Graph Neural Networks.**   Machine Learning on graphs is now largely done by Graph Neural Networks (GNNs) [37, 27, 5]. GNNs are deep architectures on graph that rely on the **Message-Passing** paradigm [16]: at each layer, the representation $H_i^l \in \mathbb{R}^{d_l}$ of each node $1 \leq i \leq N$, is updated by *aggregating* and *transforming* the representations of its neighbours at the previous layer $\{H_j^{l-1}\}_{j \in \mathcal{N}(i)}$, where $\mathcal{N}(i)$ is the neighborhood of $i$. In most examples, this aggregation can be represented as a *multiplication* of the node representation matrix $H^{l-1} \in \mathbb{R}^{N \times d_{l-1}}$ by a *propagation matrix* $S \in \mathbb{R}^{N \times N}$ related to the graph structure, followed by a fully connected transformation. That is, starting with initial node features $H^0$, the GNN $\Phi_\theta$ outputs after $k$ layers:

$$H^l = \sigma \left( S H^{l-1} \theta_l \right), \quad \Phi_\theta(H^0, S) = H^k, \tag{1}$$

where $\sigma$ is an activation function applied element-wise (often ReLU), $\theta_l \in \mathbb{R}^{d_{l-1} \times d_l}$ are learned parameters and $\theta = \{\theta_1, \ldots, \theta_k\}$. We emphasize here the dependency of the GNN on the propagation matrix $S$. Classical choices include mean aggregation $S = D^{-1}A$ or the normalized adjacency $S = D^{-\frac{1}{2}} A D^{-\frac{1}{2}}$, with $A$ the adjacency matrix of the graph and $D$ the diagonal matrix of degrees. When adding self-loops to $A$, the latter corresponds for instance to the classical GCNconv layer [27].

38th Conference on Neural Information Processing Systems (NeurIPS 2024).

An interesting example is the Simplified Graph Convolution (SGC) model [42], which consists in removing all the non-linearity ($\sigma = id$ the identity function). Surprisingly, the authors of [42] have shown that SGC reaches quite good performances when compared to non-linear architectures and due to its simplicity, SGC has been extensively employed in theoretical analyses of GNNs [46, 26].

**Graph coarsening and GNNs.** In this paper, we consider graph coarsening as a **preprocessing** step to downstream tasks [11, 21]: indeed, applying GNNs on coarsened graphs leads to drastic savings in time and memory, both during training and inference. Additionally, large graphs may be too big to fit on GPUs, and mini-batching graph nodes is known to be a difficult graph sampling problem [14], which may no longer be required on a coarsened graph. A primary question is then the following: **is training a GNN on a coarsened graph provably close to training it on the original graph?** To examine this, one must study the interaction between graph coarsening and message-passing.

There are many ways of measuring the quality of graph coarsening algorithms, following different criteria [10, 32, 7]. A classical objective is the preservation of *spectral properties* of the graph Laplacian, which gave birth to different algorithms [32, 8, 4, 24, 33]. Loukas [32] materializes this by the so-called *Restricted Spectral Approximation* (RSA, see Sec. 2) property: it roughly states that the frequency content of a certain subspace of graph signals is approximately preserved by the coarsening, or intuitively, that the coarsening is well-aligned with the low-frequencies of the Laplacian. Surprisingly, the RSA *does not generally lead to guarantees on the message-passing process* at the core of GNNs, even for very simple signals. That is, simply performing message-passing on the coarsened graph using $S_c$, the naive propagation matrix corresponding to $S$ on the coarsened graph (e.g. normalized adjacency of the coarsened graph when $S$ is the normalized adjacency of the original one) does *not* guarantee that the outputs of the GNN on the coarsened graph and the original graph will be close, even with high-quality RSA.

**Contribution.** In this paper, we address this problem by defining a **new propagation matrix** $S_c^{\mathrm{MP}}$ *specific to coarsened graphs*, which translate the RSA bound to message-passing guarantees: we show in Sec. 3.3 that training a GNN on the coarsened graph using $S_c^{\mathrm{MP}}$ is provably close to training it on the original graph. The proposed matrix $S_c^{\mathrm{MP}}$ can be computed for any given coarsening and **is not specific to the coarsening algorithm used to produce it**[1], as long as it produces coarsenings with RSA guarantees. Interestingly, our proposed matrix $S_c^{\mathrm{MP}}$ is *not symmetric* in general even when $S$ is, meaning that our guarantees are obtained by performing *oriented message-passing* on the coarsened graph, even when the original graph is undirected. To our knowledge, the only previous work to propose a new propagation matrix for coarsened graphs is [21], where the authors obtain guarantees for a specific GNN model (APPNP [28]), which is quite different from generic message-passing.

**Related Work.** Graph Coarsening originates from the multigrid-literature [36], and is part of a family of methods commonly referred to as *graph reduction*, which includes graph sampling [19], which consists in sampling nodes to extract a subgraph; graph sparsification [38, 1, 31], that focuses on eliminating edges; or more recently graph distillation [22, 45, 23], which extends some of these principles by authorizing additional informations inspired by dataset distillation [41].

Some of the first coarsening algorithms were linked to the graph clustering community, e.g. [9] which used recursively the Graclus algorithm [10] algorithm itself built on Metis [25]. Linear algebra technics such as the Kron reduction were also employed [32] [12]. In [32], the author presents a greedy algorithm that recursively merge nodes by optimizing some cost, with the purpose of preserving spectral properties of the coarsened Laplacian. This is the approach we use in our experiments (Sec. 4). It was followed by several similar methods with the same spectral criterion [8, 4, 24, 33]. Since modern graph often includes node features, other approaches proposed to take them into account in the coarsening process, often by learning the coarsening with specific regularized loss [29, 34]. Closer to this work, [11] proposes an optimization process to explicitly preserve the propagated features, however with no theoretical guarantees and only one step of message-passing. While these works often seek to preserve a fixed number of node features as in e.g. [29]), the RSA guarantees [32] leveraged in this paper are *uniform* over a whole subspace: this stronger property is required to provide guarantees for GNNs with several layers.

---

[1]Note however that $S_c^{\mathrm{MP}}$ must be computed *during the coarsening process* and included as an output of the coarsening algorithm, before eventually discarding the original graph.

Graph coarsening has been intertwined with GNNs in different ways. It can serve as graph *pooling* [44] within the GNN itself, with the aim of mimicking the pooling process in deep convolutional models on images. In the recent literature, the terms "coarsening" and "pooling" tend to be a bit exchangeable. For instance, some procedures that were initially introduced as pooling could also be used as pre-processing step, such as Graclus [10], introduced by [9] as a pooling scheme, see also [17]. Graph pooling is often data-driven and fully differentiable, such as Diffpool [44], SAGPool [30], and DMoN [39]. Theoretical work on their ability to distinguish non homomorphic graphs after pooling have been conducted [3]. In return, GNNs can also be trained to *produce* data-driven coarsenings, e.g. GOREN [6] which proposes to learn new edge weights with a GNN. As mentioned before, in the framework we consider here, graph coarsening is a *preprocessing* step with the aim of saving time and memory during training and inference [21]. Here few works derive theoretical guarantees for GNNs and message-passing, beyond the APPNP architecture examined in [21]. To our knowledge, the proposed $S_c^{\mathrm{MP}}$ is the first to yield such guarantees.

**Outline.** We start with some preliminary material on graph coarsening and spectral preservation in Sec. 2. We then present our main contribution in Sec. 3: a new propagation matrix on coarsened graphs that leads to guarantees for message-passing. As is often the case in GNN theoretical analysis, our results mostly hold for the linear SGC model, however we still outline sufficient assumptions that would be required to apply our results to general GNNs, which represent a major path for future work. In Sec. 4 we test the proposed propagation matrix on real and synthetic data, and show how it leads to improved results compared to previous works. The code is available at `https://gitlab.inria.fr/anjoly/mp-guarantees-graph-coarsening`, and proofs are deferred to App. A.

**Notations.** For a matrix $Q \in \mathbb{R}^{n \times N}$, its pseudo-inverse $Q^+ \in \mathbb{R}^{N \times n}$ is obtained by replacing its nonzero singular values by their inverse and transposing. For a symmetric positive semi-definite (p.s.d.) matrix $L \in \mathbb{R}^{N \times N}$, we define $L^{\frac{1}{2}}$ by replacing its eigenvalues by their square roots, and $L^{-\frac{1}{2}} = (L^+)^{\frac{1}{2}}$. For $x \in \mathbb{R}^N$ we denote by $\|x\|_L = \sqrt{x^\top L x}$ the Mahalanobis semi-norm associated to $L$. For a matrix $P \in \mathbb{R}^{N \times N}$, we denote by $\|P\| = \max_{\|x\|=1} \|Px\|$ the operator norm of $P$, and $\|P\|_L = \|L^{\frac{1}{2}} P L^{-\frac{1}{2}}\|$. For a subspace $R$, we say that a matrix $P$ is $R$-preserving if $x \in R$ implies $Px \in R$. Finally, for a matrix $X \in \mathbb{R}^{N \times d}$, we denote its columns by $X_{:,i}$, and define $\|X\|_{:,L} = \sum_i \|X_{:,i}\|_L$.

## 2 Background on Graph Coarsening

We mostly adopt the framework of Loukas [32], with some generalizations. A graph $G$ with $N$ nodes is described by its weighted adjacency matrix $A \in \mathbb{R}^{N \times N}$. We denote by $L \in \mathbb{R}^{N \times N}$ a notion of symmetric p.s.d. Laplacian of the graph: classical choices include the combinatorial Laplacian $L = D - A$ with $D = D(A) := \mathrm{diag}(A 1_n)$ the diagonal matrix of the degrees, or the symmetric normalized Laplacian $L = I_N - D^{-\frac{1}{2}} A D^{-\frac{1}{2}}$. We denote by $\lambda_{\max}, \lambda_{\min}$ respectively the largest and smallest non-zero eigenvalue of $L$.

**Coarsening matrix.** A coarsening algorithm takes a graph $G$ with $N$ nodes, and produces a coarsened graph $G_c$ with $n < N$ nodes. Intuitively, nodes in $G$ are grouped in "super-nodes" in $G_c$ (Fig. 1), with some weights to outline their relative importance. This mapping can be represented *via* a **coarsening matrix** $Q \in \mathbb{R}^{n \times N}$:

$$Q = \begin{cases} Q_{ki} > 0 & \text{if the } i\text{-th node of } G \text{ is mapped to the } k\text{-th super-node of } G_c \\ Q_{ki} = 0 & \text{otherwise} \end{cases}$$

The **lifting matrix** is the pseudo-inverse of the coarsening matrix $Q^+$, and plays the role of inverse mapping from the coarsened graph to the original one. The **coarsening ratio** is defined as $r$: $r = 1 - \frac{n}{N}$. That is, the higher $r$ is, the *more* coarsened the graph is.

A coarsening is said to be **well-mapped** if nodes in $G$ are mapped to a unique node in $G_c$, that is, if $Q$ has exactly one non-zero value per column. Moreover, it is **surjective** if at least one node is mapped to each super node: $\sum_i Q_{ki} > 0$ for all $k$. In this case, $QQ^\top$ is invertible diagonal and $Q^+ = Q^\top (QQ^\top)^{-1}$. Moreover, such a coarsening is said to be **uniform** when mapping weights

are constant for each super-nodes and sum to one: $Q_{ki} = 1/n_k$ for all $Q_{ki} > 0$, where $n_k$ is the number of nodes mapped to the super-node $k$. In this case the lifting matrix is particularly simple: $Q^+ \in \{0, 1\}^{N \times n}$ (Fig. 1d). For simplicity, following the majority of the literature [32], in this paper we consider only well-mapped surjective coarsenings (but not necessarily uniform).

**Remark 1.** *Non-well-mapped coarsenings may appear in the literature, e.g. when learning the matrix Q* via *a gradient-based optimization algorithm such as Diffpool [44]. However, these methods often include* regularization penalties *to favor well-mapped coarsenings.*

**Restricted Spectral Approximation.** A large part of the graph coarsening literature measures the quality of a coarsening by quantifying the modification of the spectral properties of the graph, often represented by the graph Laplacian $L$. In [32], this is done by establishing a near-isometry property for graph signals with respect to the norm $\| \cdot \|_L$, which can be interpreted as a measure of the smoothness of a signal across the graph edges. Given a signal $x \in \mathbb{R}^N$ over the nodes of $G$, we define the coarsened signal $x_c \in \mathbb{R}^n$ and the re-lifted signal $\tilde{x} \in \mathbb{R}^N$ by

$$x_c = Qx, \qquad \tilde{x} = Q^+ x_c = \Pi x \tag{2}$$

where $\Pi = Q^+ Q$. Loukas [32] then introduces the notion of *Restricted Spectral Approximation* (RSA) of a coarsening algorithm, which measures how much the projection $\Pi$ is close to the identity for a class of signals. Since $\Pi$ is at most of rank $n < N$, this cannot be true for all signals, but only for a restricted subspace $\mathcal{R} \subset \mathbb{R}^N$. With this in mind, the *RSA constant* is defined as follows.

**Definition 1** (Restricted Spectral Approximation constant). *Consider a subspace $\mathcal{R} \subset \mathbb{R}^N$, a Laplacian L, a coarsening matrix Q and its corresponding projection operator $\Pi = Q^+ Q$. The* RSA *constant $\epsilon_{L,Q,\mathcal{R}}$ is defined as*

$$\epsilon_{L,Q,\mathcal{R}} = \sup_{x \in \mathcal{R}, \|x\|_L = 1} \|x - \Pi x\|_L \tag{3}$$

In other words, the RSA constant measures how much signals in $\mathcal{R}$ are preserved by the coarsening-lifting operation, with respect to the norm $\| \cdot \|_L$. Given some $\mathcal{R}$ and Laplacian $L$, the goal of a coarsening algorithm is then to produce a coarsening $Q$ *with the smallest RSA constant possible*. While the "best" coarsening $\arg\min_Q \epsilon_{L,Q,\mathcal{R}}$ is generally computationally unreachable, there are many possible heuristic algorithms, often based on greedy merging of nodes. In App. B.1, we give an example of such an algorithm, adapted from [32]. In practice, $\mathcal{R}$ is often chosen as the subspace spanned by the first eigenvectors of $L$ ordered by increasing eigenvalue: intuitively, coarsening the graph and merging nodes is more likely to preserve the low-frequencies of the Laplacian.

While $\epsilon_{L,Q,\mathcal{R}} \ll 1$ is required to obtain meaningful guarantees, we remark that $\epsilon_{L,Q,\mathcal{R}}$ is not necessarily finite. Indeed, as $\| \cdot \|_L$ may only be a semi-norm, its unit ball is not necessarily compact. It is nevertheless finite in the following case.

**Proposition 1.** *When $\Pi$ is $\ker(L)$-preserving, it holds that $\epsilon_{L,Q,\mathcal{R}} \leq \sqrt{\lambda_{\max}/\lambda_{\min}}$.*

Hence, some examples where $\epsilon_{L,Q,\mathcal{R}}$ is finite include:

**Example 1.** *For uniform coarsenings with $L = D - A$ and connected graph $G$, $\ker(L)$ is the constant vector[2], and $\Pi$ is $\ker(L)$-preserving. This is the case examined by [32].*

**Example 2.** *For positive definite "Laplacians", $\ker(L) = \{0\}$. This is a deceptively simple solution for which $\| \cdot \|_L$ is a true norm. This can be obtained e.g. with $L = \delta I_N + \hat{L}$ for any p.s.d. Laplacian $\hat{L}$ and small constant $\delta > 0$. This leaves its eigenvectors unchanged and add $\delta$ to its eigenvalues, and therefore does not alter the fundamental structure of the coarsening problem.*

Given the adjacency matrix $A \in \mathbb{R}^{N \times N}$ of $G$, there are several possibilities to define an adjacency matrix $A_c$ for the graph $G_c$ [21, 29]. A natural choice that we make in this paper is

$$A_c = (Q^+)^\top A Q^+ . \tag{4}$$

In the case of a uniform coarsening, $(A_c)_{k\ell}$ equals the sum of edge weights for all edges in the original graph between all nodes mapped to the super-node $k$ and all nodes mapped to $\ell$. Moreover, we have the following property, derived from [32].

---

[2]Note that this would also work with several connected components, if no nodes from different components are mapped to the same super-node.

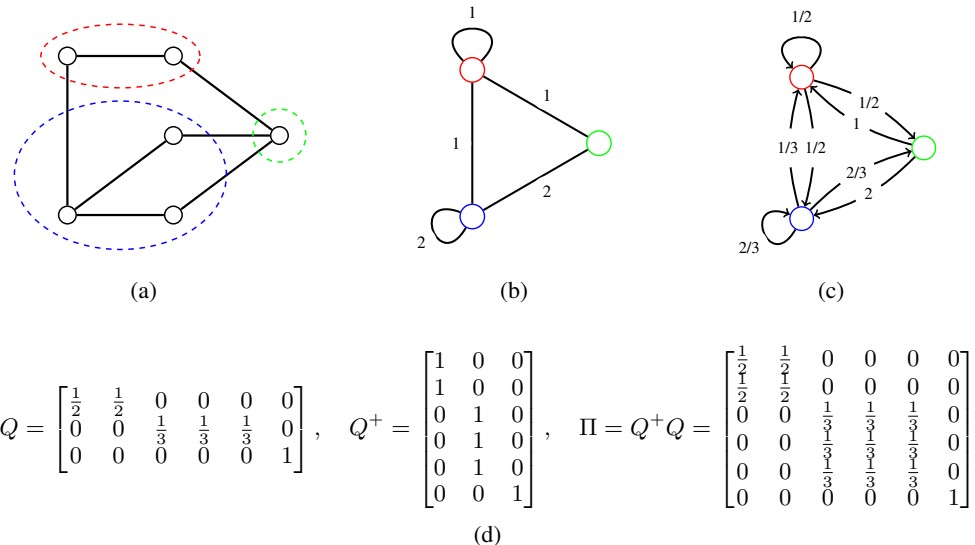

$$Q = \begin{bmatrix} \frac{1}{2} & \frac{1}{2} & 0 & 0 & 0 & 0 \\ 0 & 0 & \frac{1}{3} & \frac{1}{3} & \frac{1}{3} & 0 \\ 0 & 0 & 0 & 0 & 0 & 1 \end{bmatrix}, \quad Q^+ = \begin{bmatrix} 1 & 0 & 0 \\ 1 & 0 & 0 \\ 0 & 1 & 0 \\ 0 & 1 & 0 \\ 0 & 1 & 0 \\ 0 & 0 & 1 \end{bmatrix}, \quad \Pi = Q^+ Q = \begin{bmatrix} \frac{1}{2} & \frac{1}{2} & 0 & 0 & 0 & 0 \\ \frac{1}{2} & \frac{1}{2} & 0 & 0 & 0 & 0 \\ 0 & 0 & \frac{1}{3} & \frac{1}{3} & \frac{1}{3} & 0 \\ 0 & 0 & \frac{1}{3} & \frac{1}{3} & \frac{1}{3} & 0 \\ 0 & 0 & \frac{1}{3} & \frac{1}{3} & \frac{1}{3} & 0 \\ 0 & 0 & 0 & 0 & 0 & 1 \end{bmatrix}$$

(d)

Figure 1: Example of uniform coarsening. **(a)**: original graph $G$; **(b)**: coarsened adjacency matrix $A_c$; **(c)** representation of the proposed $S_c^{\text{MP}}$ when $S = A$; **(d)**: corresponding matrices $Q, Q^+, \Pi$.

**Proposition 2.** *Assume that the coarsening $Q$ is uniform and consider combinatorial Laplacians $L = D(A) - A$ and $L_c = D(A_c) - A_c$. Then $L_c = (Q^+)^\top L Q^+$, and*

$$\forall x \in \mathcal{R}, \quad (1 - \epsilon_{L,Q,\mathcal{R}}) \|x\|_L \le \|x_c\|_{L_c} \le (1 + \epsilon_{L,Q,\mathcal{R}}) \|x\|_L \tag{5}$$

This draws a link between the RSA and a notion of near-isometric embedding for vectors in $\mathcal{R}$. Note that the proposition above is *not* necessarily true when considering normalized Laplacians, or non uniform coarsenings. In the next section, we propose a new propagation matrix on coarsened graphs and draw a link between the RSA constant $\epsilon_{L,Q,\mathcal{R}}$ and message-passing guarantees.

## 3 Message-Passing on coarsened graphs

In the previous section, we have seen that coarsenings algorithms generally aim at preserving the spectral properties of the graph Laplacian, leading to small RSA constants $\epsilon_{L,Q,\mathcal{R}}$. However, this generally does not directly translate to guarantees on the Message-Passing process that is at the core of GNNs, which as mentioned in the introduction is materialized by the matrix $S$. In this section, we propose a new propagation matrix such that **small RSA constants leads to preserved message-passing**, which then leads to guarantees for training GNNs on coarsened graphs.

### 3.1 A new propagation matrix on coarsened graph

Given a graph $G$ with a propagation matrix $S$ and a coarsened graph $G_c$ with a coarsening matrix $Q$, our goal is to define a propagation matrix $S_c^{\text{MP}} \in \mathbb{R}^{n \times n}$ such that one round of message-passing on the coarsened signal $x_c$ followed by lifting is close to performing message-passing in the original graph: $Q^+ S_c^{\text{MP}} x_c \approx S x$. Assuming that the propagation matrix $S = f_S(A)$ is the output of a function $f_S$ of the graph's adjacency matrix, the most natural choice, often adopted in the literature [11], is therefore to simply take $S_c = f_S(A_c)$, where $A_c$ is the adjacency matrix of the coarsened graph defined in (4). However, this does not generally leads to the desired guarantees: indeed, considering for instance $S = A$, we have in this case $Q^+ S_c x_c = Q^+ (Q^+)^\top A \Pi x$, which involves the quite unnatural term $Q^+ (Q^+)^\top$. For other choices of normalized $S$, the situation is even less clear. Some authors propose different variant of $S_c$ adapted to specific cases [21, 44] (see Sec. 4), but none offers generic message-passing guarantees. To address this, we propose a new propagation matrix:

$$S_c^{\text{MP}} = Q S Q^+ \in \mathbb{R}^{n \times n}. \tag{6}$$

This expression is conceptually simple: it often amounts to some reweighting. For instance, when $S = A$ and in the case of uniform coarsening, we have $(S_c^{\text{MP}})_{k\ell} = (A_c)_{k\ell}/n_k$ (Fig. 1c). Despite

this simplicity, we will see that under some mild hypotheses this choice indeed leads to preservation guarantees of message-passing for coarsenings with small RSA constants.

**Orientation.** An important remark is that, unlike all the examples in the literature, and unlike the adjacency matrix $A_c$ defined in (4), the proposed matrix $S_c^{\text{MP}}$ **is generally asymmetric, even when $S$ is symmetric**. This means that our guarantees are obtained by performing *directed* message-passing on the coarsened graph, even when the original message-passing procedure was undirected. Conceptually, this is an important departure from previous works. However $S_c^{\text{MP}}$ becomes "more" symmetric when $Q^+$ and $Q^T$ becomes more similar. This is for instance the case when $Q$ induces a balanced partition, where each supernodes has the same number of ancestors (which can be targeted by some pooling algorithms). On the contrary, the difference between $Q$ and $Q^+$ is more pronounced when supernodes are of very different sizes,which may happen for highly irregular graphs.

## 3.2 Message-Passing guarantees

In this section, we show how the proposed propagation matrix (6) allows to transfer the spectral approximation guarantees to message-passing guarantees, under some hypotheses. First, we must make some technical assumptions relating to the kernel of the Laplacian.

**Assumption 1.** *Assume that $\Pi$ and $S$ are both $\ker(L)$-preserving.*

Moreover, since spectral approximation pertains to a subspace $\mathcal{R}$, we must assume that this subspace is left unchanged by the application of $S$.

**Assumption 2.** *Assume that $S$ is $\mathcal{R}$-preserving.*

As mentioned before, for Examples 1 and 2, the projection $\Pi$ is $\ker(L)$-preserving. Moreover, $\mathcal{R}$ is often chosen to be the subspace spanned by the low-frequency eigenvectors of $L$ and in this case, all matrices of the form $S = \alpha I_N + \beta L$ for some constant $\alpha, \beta$ are both $\ker(L)$-preserving and $\mathcal{R}$-preserving. Hence, for instance, a primary example in practice is to choose GCNconv [27] with $S = D(\hat{A})^{-\frac{1}{2}} \hat{A} D(\hat{A})^{-\frac{1}{2}}$ with $\hat{A} = A + I_N$, and to compute a coarsening with a good RSA constant for the "Laplacian" $L = (1 + \delta) I_N - S$ with small $\delta > 0$ and $\mathcal{R}$ spanned by eigenvectors of $L$. In this case, Assumptions 1 and 2 are satisfied. This is the implementation we choose in our experiments.

We now state the main result of this section.

**Theorem 1.** *Define $S_c^{\text{MP}}$ as (6). Under Assumption 1 and 2, for all $x \in \mathcal{R}$,*

$$\|Sx - Q^+ S_c^{\text{MP}} x_c\|_L \leq \epsilon_{L,Q,\mathcal{R}} \|x\|_L (C_S + C_\Pi) \tag{7}$$

*where $C_S := \|S\|_L$ and $C_\Pi := \|\Pi S\|_L$.*

*Sketch of proof.* The Theorem is proved in App. A. The proof is quite direct, and relies on the fact that, for this well-designed choice (6) of $S_c^{\text{MP}}$, the lifted signal is precisely $Q^+ S_c^{\text{MP}} x_c = \Pi S \Pi x$. Then, bounding the error incurred by $\Pi$ using the RSA, we show that this is indeed close to performing message-passing by $S$ in the original graph. $\square$

This theorem shows that the RSA error $\epsilon_{L,Q,\mathcal{R}}$ directly translates to an error bound between $Sx$ and $Q^+ S_c^{\text{MP}} x_c$. As we will see in the next section, this leads to guarantees when training a GNN on the original graph and the coarsened graph. First, we discuss the two main multiplicative constant involved in Thm. 1.

**Multiplicative constants.** In full generality, for any matrix $M$ we have $\|M\|_L \leq \|M\| \sqrt{\lambda_{\max}/\lambda_{\min}}$. Moreover, when $M$ and $L$ commute, we have $\|M\|_L \leq \|M\|$. As mentioned before, choosing $S = \alpha I_N + \beta L$ for some constants $\alpha, \beta$ is a primary example to satisfy our assumptions. In this case $C_S = \|S\|_L \leq \|S\|$. Then, if $S$ is properly normalized, e.g. for the GCNconv [27] example outlined above, we have $\|S\| \leq 1$. For combinatorial Laplacian $L = D - A$ however, we obtain $\|S\| \leq |\alpha| + |\beta| N$. We observed in our experiments that the combinatorial Laplacian generally yields poor results for GNNs.

For $C_\Pi$, in full generality we only have $C_\Pi \leq C_S \|\Pi\|_L \leq C_S \sqrt{\frac{\lambda_{\max}}{\lambda_{\min}}}$, since $\Pi$ is an orthogonal projector. However, in practice we generally observe that the exact value $C_\Pi = \|\Pi S\|_L$ is far better

than this ratio of eigenvalues (e.g. we observe a ratio of roughly $C_\Pi \approx (1/10) \cdot \sqrt{\lambda_{\max}/\lambda_{\min}}$ in our experiments). Future work may examine more precise bounds in different contexts.

### 3.3 GNN training on coarsened graph

In this section, we instantiate our message-passing guarantees to GNN training on coarsened graph, with SGC as a primary example. To fix ideas, we consider a single large graph $G$, and a node-level task such as node classification or regression. Given some node features $X \in \mathbb{R}^{N \times d}$, the goal is to minimize a loss function $J : \mathbb{R}^N \to \mathbb{R}_+$ on the output of a GNN $\Phi_\theta(X, S) \in \mathbb{R}^N$ (assumed unidimensional for simplicity) with respect to the parameter $\theta$:

$$\min_{\theta \in \Theta} R(\theta) \text{ with } R(\theta) := J(\Phi_\theta(X, S)) \tag{8}$$

where $\Theta$ is a set of parameters that we assume bounded. For instance, $J$ can be the cross-entropy between the output of the GNN and some labels on training nodes for classification, or the Mean Square Error for regression. The loss is generally minimized by first-order optimization methods on $\theta$, which requires multiple calls to the GNN on the graph $G$. Roughly, the computational complexity of this approach is $O(T(N + E)D)$, where $T$ is the number of iterations of the optimization algorithm, $D$ is the number of parameters in the GNN, and $E$ is the number of nonzero elements in $S$. Instead, one may want to train on the coarsened graph $G_c$, which can be done by minimizing instead[3]:

$$R_c(\theta) := J(Q^+ \Phi_\theta(X_c, S_c^{\mathrm{MP}})) \tag{9}$$

where $X_c = QX$. That is, the GNN is applied on the coarsened graph, and the output is then lifted to compute the loss, which is then back-propagated to compute the gradient of $\theta$. The computational complexity then becomes $O(T(n + e)D + TN)$, where $e \leq E$ is the number of nonzeros in $S_c^{\mathrm{MP}}$, and the term $TN$ is due to the lifting. As this decorrelates $N$ and $D$, it is in general much less costly.

We make the following two assumptions to state our result. Since our bounds are expressed in terms of $\| \cdot \|_L$, we must handle it with the following assumption.

**Assumption 3.** *We assume that there is a constant $C_J$ such that*

$$|J(x) - J(x')| \leq C_J \|x - x'\|_L \tag{10}$$

For most loss functions, it is easy to show that $|J(x) - J(x')| \lesssim \|x - x'\|$, and when $L$ is positive definite (Example 2) then $\| \cdot \| \leq \frac{1}{\sqrt{\lambda_{\min}}} \| \cdot \|_L$. Otherwise, one must handle the kernel of $L$, which may be done on a case-by-case basis of for an appropriate choice of $J$.

The second assumption relates to the activation function. It is here mostly for technical completeness, as *we do not have examples where it is satisfied beyond the identity $\sigma = id$*, which corresponds to the SGC architecture [42] often used in theoretical analyses [46, 26].

**Assumption 4.** *We assume that:*

i) *$\sigma$ is $\mathcal{R}$-preserving, that is, for all $x \in \mathcal{R}$, we have $\sigma(x) \in \mathcal{R}$. We discuss this constraint below.*

ii) *$\|\sigma(x) - \sigma(x')\|_L \leq C_\sigma \|x - x'\|_L$. Note that most activations are 1-Lipschitz w.r.t. the Euclidean norm, so this is satisfied when $L$ is positive-definite with $C_\sigma = \sqrt{\lambda_{\max}/\lambda_{\min}}$.*

iii) *$\sigma$ and $Q^+$ commute: $\sigma(Q^+ y) = Q^+ \sigma(y)$. This is satisfied for all uniform coarsenings, or when $\sigma$ is 1-positively homogeneous: $\sigma(\lambda x) = \lambda \sigma(x)$ for nonnegative $\lambda$ (e.g. ReLU).*

Item i) above means that, when $\mathcal{R}$ is spanned by low-frequency eigenvectors of the Laplacian, $\sigma$ does not induce high frequencies. In other words, we want $\sigma$ to preserve smooth signal. For now, the only example for which we can guarantee that Assumption 4 is satisfied is when $\sigma = id$ and the GNN is linear, which corresponds to the SGC architecture [42]. As is the case with many such analyses of GNNs, non-linear activations are a major path for future work. A possible study would be to consider random geometric graphs for which the eigenvectors of the Laplacian are close to explicit functions, e.g. spherical harmonics for dot-product graphs [2]. In this case, it may be possible to explicitly prove that Assumption 4 holds, but this is out-of-scope of this paper.

Our result on GNNs is the following.

---

[3]Note that we apply the GNN on the coarsened graph, but still lift its output to compute the loss on the training nodes of the original graph. Another possibility would be to also coarsen the labels to directly compute the loss on the coarsened graph [21], but this is not considered here. See App. B.2 for more discussion.

**Theorem 2.** *Under Assumptions 1-4: for all node features $X \in \mathbb{R}^{N \times d}$ such that $X_{:,i} \in \mathcal{R}$, denoting by $\theta^\star = \arg\min_{\theta \in \Theta} R(\theta)$ and $\theta_c = \arg\min_{\theta \in \Theta} R_c(\theta)$, we have*

$$R(\theta_c) - R(\theta^\star) \leq C\epsilon_{L,Q,\mathcal{R}} \|X\|_{:,L} \tag{11}$$

*with $C = 2C_J C_\sigma^k C_\Theta (C_S + C_\Pi) \sum_{l=1}^{k} \bar{C}_\Pi^{k-l} C_S^{l-1}$ where $\bar{C}_\Pi := \|\Pi S \Pi\|_L$ and $C_\Theta$ is a constant that depends on the parameter set $\Theta$.*

The proof of Thm. 2 is given in App. A.3. In this proof, to apply the Theorem 1, we apply the RSA to each nodes features column. It relies on the assumption that each column of the nodes features $X_{:,i}$ belongs to the preserved space $\mathcal{R}$. This assumption seems reasonable for homophilic datasets (Cora, Citeseer) and large preserved space . The Theorem states that training a GNN that uses the proposed $S_c^{\text{MP}}$ on the coarsened graph by minimizing (9) yields a parameter $\theta_c$ whose excess loss compared to the optimal $\theta^\star$ is bounded by the RSA constant. Hence, spectral approximation properties of the coarsening directly translates to guarantees on GNN training. The multiplicative constants $C_S, C_\Pi$ have been discussed in the previous section, and the same remarks apply to $\bar{C}_\Pi$.

## 4  Experiments

**Setup.**  We choose the propagation matrix from GCNconv [27], that is, $S = f_S(A) = D(\hat{A})^{-\frac{1}{2}} \hat{A} D(\hat{A})^{-\frac{1}{2}}$ with $\hat{A} = A + I_N$. As detailed in the previous section, we take $L = (1+\delta)I_N - S$ with $\delta = 0.001$ and $\mathcal{R}$ as the $K$ first eigenvectors of $L$ ($K = N/10$ in our experiments), ensuring that Assumptions 1 and 2 are satisfied. In our experiments, we observed that the combinatorial Laplacian $L = D - A$ gives quite poor results, as it corresponds to unusual propagation matrices $S = \alpha I_N + \beta L$, and the constant $C_S = \|S\|_L$ is very large. Hence our focus on the normalized case.

On coarsened graphs, we compare five propagation matrices:

- $S_c^{\text{MP}} = QSQ^+$, our proposed matrix
- $S_c = f_S(A_c)$, the naive choice
- $S_c^{diag} = \hat{D}'^{-1/2}(A_c + C)\hat{D}'^{-1/2}$, proposed in [21], where $C$ is the diagonal matrix of the $n_k$ and $\hat{D}'$ the corresponding degrees. This yields theoretical guarantees for APPNP when $S$ is GCNconv;
- $S_c^{diff} = QSQ^\top$, which is roughly inspired by Diffpool [44];
- $S_c^{sym} = (Q^+)^\top SQ^+$, which is the lifting employed to compute $A_c$ (4).

**Coarsening algorithm.**  Recall that the proposed $S_c^{\text{MP}}$ can be computed for any coarsening, and that the corresponding theoretical guarantees depend on the RSA constant $\epsilon_{L,Q,\mathcal{R}}$. In our experiments, we adapt the algorithm from [32] to coarsen the graphs. It takes as input the graph $G$ and the coarsening ratio desired $r$ and output the propagation matrix $S_c^{\text{MP}}$ and the coarsening matrix $Q$ used for lifting. It is a greedy algorithm which successively merges edges by minimizing a certain cost. While originally designed for the combinatorial Laplacian, we simply adapt the cost to any Laplacian $L$, see App. B.1. Note however that some mathematical justifications for this approach in [32] are no longer valid for normalized Laplacian, but we find in practice that it produces good RSA constants.

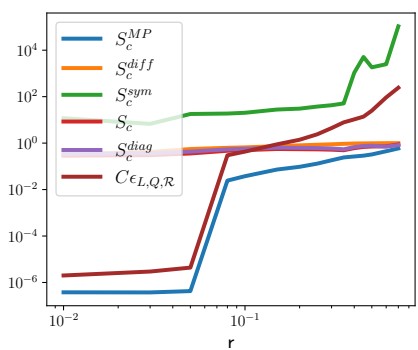

Figure 2: Message-Passing error for different propagation matrices.

A major limit of this algorithm is its computational cost, which is quite high since it involves large matrix inversion and SVD computation. Hence we limit ourselves to middle-scale graphs like Cora [35] and Citeseer [15] and one larger graph with Reddit [18] in the following experiments. The design of more scalable coarsening algorithms with RSA guarantees is an important path for future work, but out-of-scope of this paper.

**Message passing preservation guarantees** To evaluate the effectiveness of the proposed propagation matrix, we first illustrate the theoretical message passing preservation guarantees (Thm. 1 and 2) on synthetic graphs, taken as random geometric graph, built by sampling 1000 nodes with coordinates in $[0,1]^2$ and connecting them if their distance is under a given threshold (details in App. B.3). For each choice of propagation matrix and different coarsening ratio, we compute numerically $\|S^k x - Q^+(S_c^{\mathrm{MP}})^k x_c\|_L$ for various signals $x \in \mathcal{R}$. We perform $N_p = 6$ message-passing steps to enhance the difference between propagation matrices. We evaluate and plot the upper bound defined by $\epsilon_{L,Q,\mathcal{R}}(C_S + C_\Pi) \sum_{l=1}^{k} \bar{C}_\Pi^{k-l} C_S^{l-1}$ (seen in the proof of Theorem 2 in App. A.3) in Fig. 2. We observe that our propagation matrix incurs a significantly lower error compared to other choices, and that as expected, this error is correlated to $\epsilon_{L,Q,\mathcal{R}}$, which is not the case for other choices. More experiments can be found in App. B.4.

**Node classification on real graphs.** We then perform node classification experiments on real-world graphs, namely Cora [35] and Citeseer [15], using the public split from [43]. For simplicity, we restrict them to their largest connected component[4], since using a connected graph is far more convenient for coarsening algorithms (details in App. B.3). The training procedure follows that of Sec. 3.3: the network is applied to the coarsened graph and coarsened node features, its output is lifted to the original graph with $Q^+$, and the label of the original training graph nodes are used to compute the cross-entropy loss, which is then back-propagated to optimize the parameters $\theta$ (pseudocode in App. B.2). Despite the lifting procedure, this results in faster training than using the entire graph (e.g., by approximately $30\%$ for a coarsening ratio of $r = 0.5$ when parallelized on GPU). For downstream tasks we introduce a novel metric to analyze a specific coarsening : "Max acc possible". It corresponds to the optimal prediction over the super-nodes of the coarsened graph (all the nodes coarsened in a super nodes has the same prediction, optimally the majority label of this cluster). It might be hard to achieve as the optimal assignment for the validation nodes or training nodes can be different. It allows comparing different coarsenings for classification task without training models on it. We test SGC [42] with $N_p = 6$ and GCNconv [27] with $N_p = 2$ on four different coarsening ratio: $r \in \{0.3, 0.5, 0.7\}$ where $N_p$ is the number of propagation. Each classification results is averaged on 10 random training.

Results are reported in Table 1 and Table 2. We observe that the proposed propagation matrix $S_c^{\mathrm{MP}}$ yields better results and is more stable, especially for high coarsening ratio. The benefits are more evident when applied to the SGC architecture [42], for which Assumption 4 of Thm. 2 is actually satisfied, than for GCN, for which ReLU is unlikely to satisfy Assumption 4. It is also interesting to notice that training on coarsened graphs sometimes achieve better results than on the original graph. This may be explained by the fact that, for homophilic graphs (connected nodes are more likely to have the same label), nodes with similar labels are more likely to be merged together during the coarsening, and thus become easier to predict for the model. The detailed hyper-parameters for each model and each dataset can be found in appendix B.5.

Table 1: Accuracy in $\%$ for node classification with SGC and different coarsening ratio

| SGC | Cora | | | Citeseer | | |
|---|---|---|---|---|---|---|
| $r$ | 0.3 | 0.5 | 0.7 | 0.3 | 0.5 | 0.7 |
| $S_c^{sym}$ | $16.8 \pm 3.8$ | $16.1 \pm 3.8$ | $16.4 \pm 4.7$ | $17.5 \pm 3.8$ | $18.6 \pm 4.6$ | $19.8 \pm 5.0$ |
| $S_c^{diff}$ | $50.7 \pm 1.4$ | $21.8 \pm 2.2$ | $13.6 \pm 2.8$ | $50.5 \pm 0.2$ | $30.5 \pm 0.2$ | $23.1 \pm 0.0$ |
| $S_c$ | $79.3 \pm 0.1$ | $78.7 \pm 0.0$ | $74.6 \pm 0.1$ | $\mathbf{74.1} \pm 0.1$ | $72.8 \pm 0.1$ | $72.5 \pm 0.1$ |
| $S_c^{diag}$ | $79.9 \pm 0.1$ | $78.7 \pm 0.1$ | $77.3 \pm 0.0$ | $73.6 \pm 0.1$ | $73.4 \pm 0.1$ | $73.1 \pm 0.4$ |
| $S_c^{\mathrm{MP}}$ **(ours)** | $\mathbf{81.8} \pm 0.1$ | $\mathbf{80.3} \pm 0.1$ | $\mathbf{78.5} \pm 0.0$ | $73.9 \pm 0.1$ | $\mathbf{74.6} \pm 0.1$ | $\mathbf{74.2} \pm 0.1$ |
| Max acc possible | 96.5 | 92.5 | 88.9 | 93.5 | 90.5 | 84.5 |
| Full Graph | | $81.6 \pm 0.1$ | | | $73.6 \pm 0.0$ | |

**Scaling to larger Datasets** We performed experiments on the Reddit Dataset [18], which is approximately 100 times bigger than Cora or Citeseer. The Message-Passing error for different coarsening propagation matrices is reported in Table 3 with the node prediction results on two coarsening ratio $r = 90\%$ and $r = 99\%$ (their number of nodes,and edges can be found in App B.3), the details of the hyperparameters and coarsening procedure are in B.6. Our propagation matrix for

---

[4]hence the slight difference with other reported results on these datasets

Table 2: Accuracy in % for node classification with GCNconv and different coarsening ratio

| GCNconv | Cora | | | Citeseer | | |
|---|---|---|---|---|---|---|
| $r$ | 0.3 | 0.5 | 0.7 | 0.3 | 0.5 | 0.7 |
| $S_c^{sym}$ | $80.1 \pm 1.3$ | $78.1 \pm 1.3$ | $30.8 \pm 2.5$ | $71.0 \pm 1.4$ | $62.5 \pm 11$ | $52.7 \pm 3.6$ |
| $S_c^{diff}$ | $81.9 \pm 1.0$ | $74.5 \pm 0.9$ | $62.6 \pm 7.1$ | $72.7 \pm 0.4$ | $71.2 \pm 1.7$ | $37.6 \pm 0.9$ |
| $S_c$ | $81.2 \pm 0.8$ | $79.9 \pm 0.9$ | $78.1 \pm 1.0$ | $71.7 \pm 0.6$ | $70.7 \pm 1.0$ | $67.1 \pm 3.1$ |
| $S_c^{diag}$ | $81.4 \pm 0.8$ | $\mathbf{80.4} \pm 0.8$ | $\mathbf{78.6} \pm 1.3$ | $72.1 \pm 0.6$ | $70.2 \pm 0.8$ | $69.3 \pm 1.9$ |
| $S_c^{MP}$ (ours) | $\mathbf{82.1} \pm 0.5$ | $79.8 \pm 1.5$ | $78.2 \pm 0.9$ | $\mathbf{72.8} \pm 0.8$ | $\mathbf{72.0} \pm 0.8$ | $\mathbf{70.0} \pm 1.0$ |
| Max acc possible | 96.5 | 92.5 | 88.9 | 93.5 | 90.5 | 84.5 |
| Full Graph | | $81.6 \pm 0.6$ | | | $73.1 \pm 1.5$ | |

coarsened graphs achieved a better Message-Passing error, close to the RSA-constant computed in the coarsened graph. It is consistent with the fact the Message-Passing error is bounded by Theorem 1 with our propagation matrix. Similarly, for the node prediction results, our propagation matrix $S_c^{MP}$ achieves good results with the SGC model, close to the maximum accuracy possible on the given coarsening. Our propagation matrix is still competitive with the GCNconv model and achieved better results on the biggest coarsening ratio. These experiments show the effectiveness of our method on large graphs for which coarsening as a preprocessing step is crucial: indeed, on most small-scale machines with single GPU, the Reddit dataset is too large to fit in memory and requires adapted strategies.

Table 3: Accuracy in % for node classification on Reddit Dataset and Message passing errors

| Reddit Dataset | SGC | | GCNconv | | MP error | |
|---|---|---|---|---|---|---|
| $r$ | 0.90 | 0.99 | 0.90 | 0.99 | 0.90 | 0.99 |
| $S_c^{sym}$ | $37.1 \pm 6.6$ | $3.7 \pm 5.5$ | $48.1 \pm 8.9$ | $34.8 \pm 4.0$ | 4.73e16 | 2.07e27 |
| $S_c^{diff}$ | $18.3 \pm 0.0$ | $14.9 \pm 0.0$ | $71.3 \pm 1.0$ | $18.7 \pm 1.7$ | 0.92 | 1.00 |
| $S_c$ | $87.5 \pm 0.1$ | $37.3 \pm 0.0$ | $88.0 \pm 0.1$ | $54.2 \pm 2.4$ | 2.46 | 1.75 |
| $S_c^{diag}$ | $87.6 \pm 0.1$ | $37.3 \pm 0.0$ | $\mathbf{88.1} \pm 0.2$ | $55.5 \pm 1.8$ | 2.45 | 1.74 |
| $S_c^{MP}$ (ours) | $\mathbf{90.2} \pm 0.0$ | $\mathbf{64.1} \pm 0.0$ | $84.4 \pm 0.3$ | $\mathbf{60.3} \pm 0.9$ | $\mathbf{0.22}$ | $\mathbf{0.88}$ |
| Max Acc Possible | 93.4 | 64.7 | 93.4 | 64.7 | Not applicable | |
| Full Graph | | 94.9 | Non computable (OOM) | | Not applicable | |

## 5 Conclusion

In this paper, we investigated the interactions between graph coarsening and Message-Passing for GNNs. Surprisingly, we found out that even for high-quality coarsenings with strong spectral preservation guarantees, naive (but natural) choices for the propagation matrix on coarsened graphs does not lead to guarantees with respect to message-passing on the original graph. We thus proposed a new message-passing matrix specific to coarsened graphs, which naturally translates spectral preservation to message-passing guarantees, for any coarsening, under some hypotheses relating to the structure of the Laplacian and the original propagation matrix. We then showed that such guarantees extend to GNN, and in particular to the SGC model, such that training on the coarsened graph is provably close to training on the original one.

There are many outlooks to this work. Concerning the coarsening procedure itself, which was not the focus of this paper, new coarsening algorithms could emerge from our theory, e.g. by instantiating an optimization problem with diverse regularization terms stemming from our theoretical bounds. The scalability of such coarsening algorithms is also an important topic for future work. From a theoretical point of view, a crucial point is the interaction between non-linear activation functions and the low-frequency vectors in a graph (Assumption 4). We focused on the SGC model here, but a more in-depth study of particular graph models (e.g. random geometric graphs) could shed light on this complex phenomenon, which we believe to be a major path for future work.

## Acknowledgments and Disclosure of Funding

The authors acknowledge the fundings of France 2030, PEPR IA, ANR-23-PEIA-0008 and ANR GrandMa ANR-21-CE23-0006.

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

# A Proofs

We start by a small useful lemma.

**Lemma 1.** *For all $\ker(L)$-preserving matrices $M$, we have $\|Mx\|_L \le \|M\|_L \|x\|_L$.*

*Proof.* Take the orthogonal decomposition $x = u + v$ where $u \in \ker(L)$ and $v \in \ker(L)^\perp$. Then, since $\|x\|_L = \|v\|_L$, $L^{-\frac{1}{2}} L^{\frac{1}{2}} v = v$ and $\|Mu\|_L = 0$,

$$\|Mx\|_L \le \|Mv\|_L = \|L^{\frac{1}{2}} M L^{-\frac{1}{2}} L^{\frac{1}{2}} v\| \le \|M\|_L \|v\|_L$$

$\square$

## A.1 Proof of Proposition 2

*Proof.* The fact that $L_c = (Q^+)LQ^+$ in this particular case is done by direct computation. It results that $\|x_c\|_{L_c} = \|L^{\frac{1}{2}} \Pi x\|$ and

$$\big|\, \|x\|_L - \|x_c\|_{L_c}\, \big| \le \|L^{\frac{1}{2}} x - L^{\frac{1}{2}} \Pi x\| = \|x - \Pi x\|_L \le \epsilon_{L,Q,\mathcal{R}} \|x\|_L$$

by definition of $\epsilon_{L,Q,\mathcal{R}}$. $\square$

## A.2 Proof of Theorem 1

*Proof.* Since $x \in \mathcal{R}$ and $S$ is $\mathcal{R}$-preserving, we have

$$\|\Pi^\perp x\|_L \le \epsilon_{L,Q,\mathcal{R}} \|x\|_L$$

where $\Pi^\perp = I_N - \Pi$, and similarly for $Sx$. Moreover, under Assumption 1, both $\Pi$ and $S$ are $\ker(L)$-preserving, such that $\|\Pi S x\|_L \le \|\Pi S\|_L \|x\|_L$ for all $x$. Then

$$
\begin{aligned}
\|Sx - Q^+ S_c^{\mathrm{MP}} x_c\|_L &= \|Sx - \Pi S \Pi x\|_L \\
&= \|Sx - \Pi Sx + \Pi Sx - \Pi S \Pi x\|_L \\
&= \|\Pi^\perp Sx + \Pi S \Pi^\perp x\|_L \\
&\le \|\Pi^\perp Sx\|_L + \|\Pi S \Pi^\perp x\|_L \\
&\le \epsilon_{L,Q,\mathcal{R}} \|Sx\|_L + \|\Pi S\|_L \|\Pi^\perp x\|_L \\
&\le \epsilon_{L,Q,\mathcal{R}} \|Sx\|_L + \epsilon_{L,Q,\mathcal{R}} \|\Pi S\|_L \|x\|_L = \epsilon_{L,Q,\mathcal{R}} \|x\|_L \left( C_S + C_\Pi \right)
\end{aligned}
$$

$\square$

## A.3 Proof of Theorem 2

Recall that the GNN is such that $H^0 = X$, and

$$H^l = \sigma(SH^{l-1}\theta_l) \in \mathbb{R}^{N \times d_\ell}, \quad \Phi_\theta(X, S) = H^k \in \mathbb{R}^N$$

Similarly, for the GNN on coarsened graph we denote by $H_c^0 = X_c$ and its layers

$$H_c^l = \sigma(S_c^{\mathrm{MP}} H_c^{l-1}\theta_l) \in \mathbb{R}^{n \times d_\ell}, \quad \Phi_\theta(X_c, S_c^{\mathrm{MP}}) = H_c^k \in \mathbb{R}^N$$

For some set of parameters $\theta$ of a GNN, we define

$$C_{\theta,l} = \sup_i \sum_j |\theta_{ij}^l|, \qquad \bar{C}_{\theta,l} = \prod_{p=1}^l C_{\theta,p}$$

We start with a small lemma.

**Lemma 2.** *Define*

$$B_l = B_l(X) := \sum_i \|H_{:,i}^l\|_L \tag{12}$$

*Then we have*

$$B_l \le \bar{C}_{\theta,l} C_S^l C_\sigma^l \|X\|_{:,L} \tag{13}$$

*Proof.* From assumption 4 we have $\|\sigma(x)\|_L \le C_\sigma \|x\|_L$. Then, since $S$ is $\ker(L)$-preserving from Assumption 1, by Lemma 1

$$\sum_i \|H^l_{:,i}\|_L = \sum_i \|\sigma(SH^{l-1}\theta^l_{:,i})\|_L \le C_\sigma \sum_i \|SH^{l-1}\theta^l_{:,i}\|_L$$

$$\le C_\sigma (\sup_j \sum_i |\theta^l_{ji}|) \sum_j \|SH^{l-1}_{:,j}\|_L \le C_\sigma C_{\theta,l} C_S B_{l-1}$$

Since $B_0 = \|X\|_{:,L}$, we obtain the result $\qquad\qquad\square$

*Proof.* We start with classical risk bounding in machine learning

$$J(\Phi_{\theta_c}(X,S)) - J(\Phi_{\theta^\star}(X,S)) = J(\Phi_{\theta_c}(X,S)) - J(Q^+\Phi_{\theta_c}(X_c, S_c^{\mathrm{MP}}))$$
$$+ J(Q^+\Phi_{\theta_c}(X_c, S_c^{\mathrm{MP}})) - J(Q^+\Phi_{\theta^\star}(X_c, S_c^{\mathrm{MP}}))$$
$$+ J(Q^+\Phi_{\theta^\star}(X_c, S_c^{\mathrm{MP}})) - J(\Phi_{\theta^\star}(X,S))$$
$$\le 2 \sup_{\theta \in \Theta} |J(\Phi_\theta(X,S)) - J(Q^+\Phi_\theta(X_c, S_c^{\mathrm{MP}}))|$$

since $\theta_c$ minimizes $J(Q^+\Phi_\theta(X_c, S_c^{\mathrm{MP}}))$. For all $\theta$, by Assumption 3, we have

$$|J(\Phi_\theta(X,S)) - J(Q^+\Phi_\theta(X_c, S_c^{\mathrm{MP}}))| \le C_J \|\Phi_\theta(X,S) - Q^+\Phi_\theta(X_c, S_c^{\mathrm{MP}})\|_L \qquad (14)$$

We will prove a recurrence bound on

$$E_l := \sum_i \|H^l_{:,i} - Q^+(H^l_c)_{:,i}\|_L$$

From Assumption 4,

$$E_l = \sum_i \|\sigma(SH^{l-1}(\theta_l)_{:,i}) - Q^+\sigma(S_c^{\mathrm{MP}} H_c^{l-1}(\theta_l)_{:,i})\|_L$$

$$= \sum_i \|\sigma(SH^{l-1}(\theta_l)_{:,i}) - \sigma(Q^+ S_c^{\mathrm{MP}} H_c^{l-1}(\theta_l)_{:,i})\|_L$$

$$\le C_\sigma \sum_i \|SH^{l-1}(\theta_l)_{:,i} - Q^+ S_c^{\mathrm{MP}} H_c^{l-1}(\theta_l)_{:,i}\|_L$$

$$\le C_\sigma \sum_j \left( \sum_i |(\theta_l)_{ji}| \right) \|SH^{l-1}_{:,j} - Q^+ S_c^{\mathrm{MP}}(H_c^{l-1})_{:,j}\|_L$$

We then write

$$\|SH^{l-1}_{:,j} - Q^+ S_c^{\mathrm{MP}}(H_c^{l-1})_{:,j}\|_L \le \|SH^{l-1}_{:,j} - Q^+ S_c^{\mathrm{MP}} QH^{l-1}_{:,j}\|_L + \|Q^+ S_c^{\mathrm{MP}} QH^{l-1}_{:,j} - Q^+ S_c^{\mathrm{MP}}(H_c^{l-1})_{:,j}\|_L$$

Then note that, since both $S$ and $\sigma$ are $\mathcal{R}$-preserving, for all $l, i$ we have that $(H^l)_{:,i} \in \mathcal{R}$. We can thus apply Theorem 1 to the first term:

$$\|SH^{l-1}_{:,j} - Q^+ S_c^{\mathrm{MP}} QH^{l-1}_{:,j}\|_L \le \epsilon_{L,Q,\mathcal{R}}(C_S + C_\Pi)\|H^{l-1}_{:,j}\|_L$$

The second term is $0$ when $l = 1$ since $H_c^0 = QH^0$. Otherwise, using $QQ^+ = I_n$, and since under Assumption 1 both $S$ and $\Pi$ are $\ker(L)$-preserving, applying Lemma 1:

$$\|Q^+ S_c^{\mathrm{MP}} QH^{l-1}_{:,j} - Q^+ S_c^{\mathrm{MP}}(H_c^{l-1})_{:,j}\|_L = \|Q^+ S_c^{\mathrm{MP}} Q(H^{l-1}_{:,j} - Q^+(H_c^{l-1})_{:,j})\|_L$$

$$= \|\Pi S\Pi(H^{l-1}_{:,j} - Q^+(H_c^{l-1})_{:,j})\|_L$$

$$\le \|\Pi S\Pi\|_L \|H^{l-1}_{:,j} - Q^+(H_c^{l-1})_{:,j}\|_L$$

At the end of the day, we obtain

$$E_l \le C_\sigma C_{\theta,l}(\epsilon_{L,Q,\mathcal{R}}(C_S + C_\Pi)B_{l-1} + \bar{C}_\Pi E_{l-1})$$
$$\le C_\sigma^l \bar{C}_{\theta,l} C_S^{l-1}(C_S + C_\Pi)\epsilon_{L,Q,\mathcal{R}}\|X\|_{:,L} + C_\sigma C_{\theta,l}\bar{C}_\Pi E_{l-1}$$

using Lemma 2, and $E_1 \leq \epsilon_{L,Q,\mathcal{R}} C_\sigma C_{\theta,1}(C_S + C_\Pi)\|X\|_{:,L}$. We recognize a recursion of the form $u_n \leq a_n c + b_n u_{n-1}$, which leads to $u_n \leq \sum_{p=2}^n a_p \prod_{i=p+1}^n b_i + u_1 \prod_{i=2}^n b_i$, which results in:

$$E_k \leq \epsilon_{L,Q,\mathcal{R}}\|X\|_{:,L} C_\sigma^k \bar{C}_{\theta,k}, (C_S + C_\Pi) \sum_{l=1}^k \bar{C}_\Pi^{k-l} C_S^{l-1} \tag{15}$$

This concludes the proof with $C_\Theta = \max_{\theta \in \Theta} \bar{C}_{\theta,k}$.

$\square$

## B Coarsening algorithm and experimental details

### B.1 Adaptation of Loukas Algorithm

You can find below the pseudo-code of Loukas algorithm. This algorithm works by greedy selection of *contraction sets* (see below) according to some cost, merging the corresponding nodes, and iterate. The main modification is to replace the combinatorial Laplacian in the Loukas code by any Laplacian $L = f_L(A)$, and to update the adjacency matrix according to (4) at each iteration and recompute $L$, instead of directly updating $L$ as in the combinatorial Laplacian case. Note that we also remove the diagonal of $A_c$ at each iteration, as we find that it produces better results. The output of the algorithm is the resulting coarsening $Q$, as well as $S_c^{\text{MP}} = QSQ^+$ for our application.

---

**Algorithm 1** Loukas Algorithm

---

**Require:** Adjacency matrix $A$, Laplacian $L = f_L(A)$, propagation matrix $S$, a coarsening ratio $r$, preserved space $\mathcal{R}$, maximum number of nodes merged at one coarsening step : $n_e$
1: $n_{obj} \leftarrow \text{int}(N - N \times r)$ the number of nodes wanted at the end of the algorithm.
2: compute cost matrix $B_0 \leftarrow VV^T L^{-1/2}$ with $V$ an orthonormal basis of $\mathcal{R}$
3: $Q \leftarrow I_N$
4: **while** $n \geq n_{obj}$ **do**
5:     Make one coarsening STEP $l$
6:     Create candidate contraction sets.
7:     For each contraction $\mathcal{C}$, compute $\text{cost}(\mathcal{C}, B_{l-1}, L_{l-1}) = \frac{\|\Pi_C B_{l-1}(B_{l-1}^T L_{l-1} B_{l-1})^{-1/2}\|_{L_\mathcal{C}}}{|\mathcal{C}|-1}$
8:     Sort the list of contraction set by the lowest score
9:     Select the lowest scores non overlapping contraction set while the number of nodes merged is inferior to $\min(n - n_{obj}, n_e)$
10:     Compute $Q_l$, $Q_l^+$, uniform intermediary coarsening with contraction sets selected
11:     $B_l \leftarrow Q_l B_{l-1}$
12:     $Q \leftarrow Q_l Q$
13:     $A_l \leftarrow (Q_l^+)^\top A_{l-1} Q_l^+ - \text{diag}((Q_l^+)^\top A_{l-1} Q_l^+)1_n)$
14:     $L_{l-1} = f_L(A_{l-1})$
15:     $n \leftarrow \min(n - n_{obj}, n_e)$
16: **end while**
17: IF uniform coarsening THEN $Q \leftarrow \text{row-normalize}(Q_l Q)$
18: Compute $S_c^{\text{MP}} = QSQ^+$
19: **return** $Q, S_c^{\text{MP}}$

---

The terms $\Pi_\mathcal{C}$ and $L_\mathcal{C}$ are some specific projection of the contraction set, their explicit definition can be find in Loukas work [32]. We did not modify them here and leave their eventual adaptation for future work.

**Enforcing the iterative/greedy aspect** In our adaptation we also add a parameter $n_e$ to limit the number of nodes contracted at each coarsening step. In one coarsening step, when a contraction set $\mathcal{C}$ is selected, we merge $|\mathcal{C}|$ nodes. In practice Loukas proposed in its implementation to force $n_e = \infty$ and coarsen the graph in one single iteration. We observed empirically better results by diminishing $n_e$ and combining it with enforcing the uniform coarsening (Appendix B.4).

**Candidate contraction Set.**    Candidate contractions sets come in two main flavors: they can be each two nodes linked by edges, or the neighborhood of each nodes (so-called "variation edges" and "variation neighborhood" versions). In practice, as the neighborhood are quite big in our graphs, it is not very convenient for small coarsening ratio and give generally poor results. We will use mainly the edges set as candidate contraction sets and adjust the parameter $n_e$ to control the greedy aspect of this algorithm.

**Uniform Coarsening**    At each coarsening step, in Loukas algorithm $Q_l$ is uniform by construction. Nonetheless the product of uniform coarsening is not necessarily an uniform coarsening. Then, we propose an option to force the uniform distribution in the super-nodes in the Loukas algorithm by normalize the non zero values of each line of the final coarsening matrix $Q$. We observe that uniform coarsening gives better results for $\epsilon_{L,Q,\mathcal{R}}$, and works better for our message passing guarantees. See Appendix B.4.

## B.2    Discussion on Training procedure

The pseudocode of our training procedure is detailed in Algo. 2.

---

**Algorithm 2** Training Procedure

---

**Require:** Adjacency $A$, node features $X$, desired propagation matrix $S$, preserved space $\mathcal{R}$, Laplacian $L$, a coarsening ratio $r$
 1: $Q, S_c^{\text{MP}} \leftarrow$ Coarsening-algorithm$(A, L, S, r, \mathcal{R})$
 2: $X_c \leftarrow QX$
 3: Initialize model (SGC or GCNconv)
 4: **for** $N_{epochs}$ iterations **do**
 5:     compute coarsened prediction $\Phi_\theta(S_c^{\text{MP}}, X_c)$
 6:     uplift the predictions : $Q^+\Phi_\theta(S_c^{\text{MP}}, X_c)$
 7:     compute the cross entropy loss $J(Q^+\Phi_\theta(S_c^{\text{MP}}, X_c))$
 8:     Backpropagate the gradient
 9:     Update $\theta$
10: **end for**

---

Note that it is different from the procedure of [21] which computes labels for the super-nodes (using the majority label in the coarsening cluster) and do not use the uplifting matrix $Q^+$. We find this procedure to be less amenable to semi-supervised learning, as super-nodes may merge training and testing nodes, and prefer to uplift the output of the GNN in the original graph instead. Additionally, this preserves the theoretical guarantees of Sec. 3. Our procedure might be slightly lower but we find the uplifting operation to be of negligible cost compared to actual backpropagation.

## B.3    Presentation of dataset

**Synthetic Dataset**    Random geometric graph is built by sampling nodes with coordinates in $[0, 1]^2$ and connecting them if their distance is under a given threshold. For the experiment on illustrating the message passing preservation guarantees, we sample 1000 nodes with a threshold of 0.05 (fig 3 ).

Table 4: Characteristics of Cora and CiteSeer Datasets

| Dataset | # Nodes | # Edges | # Train Nodes | # Val Nodes | # Test Nodes |
|---|---|---|---|---|---|
| Cora | 2,708 | 10,556 | 140 | 500 | 1,000 |
| Cora PCC | 2,485 | 10,138 | 122 | 459 | 915 |
| Citeseer | 3,327 | 9,104 | 120 | 500 | 1,000 |
| Citeseer PCC | 2,120 | 7,358 | 80 | 328 | 663 |

**Real World datasets**    We restrict the well known Cora and Citeseer to their principal connected component(PCC) as it more compatible with coarsening as preprocessing. Indeed, the loukas

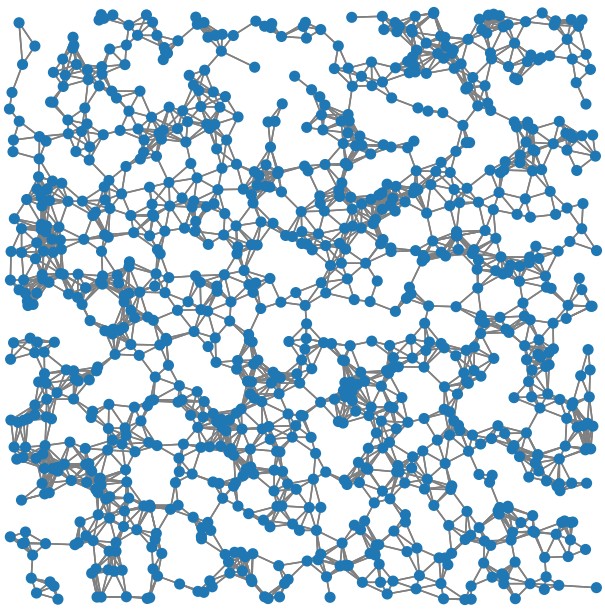

Figure 3: Example of a random Geometric graph

algorithm tend to coarsen first the smallest connected components before going to the biggest which leads to poor results for small coarsening ratio. However working with this reduced graph make the comparison with other works more difficult as it is not the same training and evaluating dataset.

For the Reddit dataset ( 1 PCC) its characteristics and of its coarsened version as well of the Reddit and Cora coarsened dataset can be find in the table 5

Table 5: Characteristics of Reddit, Cora, Citeseer and its coarsen version

| Dataset | # Nodes | # Edges | # Features | #classes |
|---|---|---|---|---|
| Reddit | 232,965 | 114,615,892 | 602 | 41 |
| Reddit90 | 23,298 | 8,642,864 | 602 | 41 |
| Reddit99 | 2,331 | 10,838 | 602 | 41 |
| Cora PCC | 2,485 | 10,138 | 1,433 | 7 |
| Cora70 | 746 | 3,716 | 1,433 | 7 |
| Citeseer PCC | 2,120 | 7,358 | 3,703 | 6 |
| Citeseer70 | 636 | 2,122 | 3,703 | 6 |

## B.4 Discussion of hyperparameters and additional experiments

In the following section, we will use two different view of the same plot, to focus on different parts. We use the log-log scale (fig 4a) to put the focus on low coarsening ratio and on the upper bound. We use the linear scale (fig 4b) to compare more precisely our propagation matrix with $S_c^{diag}$ and $S_c$ for higher coarsening ratio.

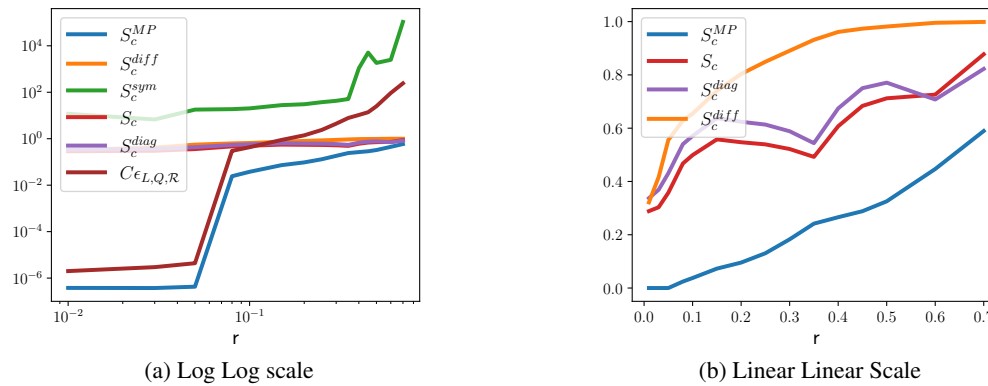

| (a) Log Log scale | (b) Linear Linear Scale |

Figure 4: Uniform coarsening with $n_e = 5N/100$ and Normalized Laplacian

**Uniform coarsening** We observe that forcing uniform coarsening gives better $\epsilon_{L,Q,\mathcal{R}}$ and thus better message passing guarantees . It is shown in the figure 5 for $n_e = 5N/100$ with N the number of Nodes of the graph (1000 here).

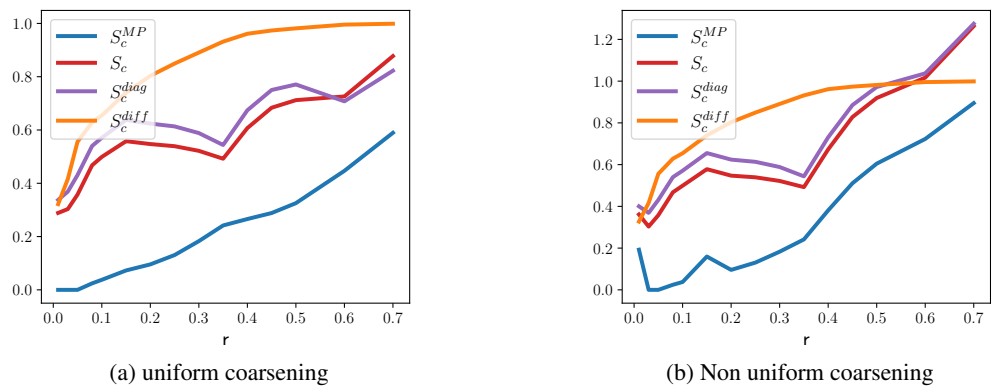

| (a) uniform coarsening | (b) Non uniform coarsening |

Figure 5: Coarsening with $n_e = 5N/100$ and Normalized Laplacian

**Bounding $n_e$.** For high coarsening ratio, we observe limits of the variation edges defined as Loukas with $n_e \to \infty$ as it gives bigger $\epsilon_{L,Q,\mathcal{R}}$ and thus worse curve for our propagation matrix in the coarsened graph (fig 6).

### B.5  Hyper-parameters for Table 1 and Table 2

For all experiments, we preserve $K$ eigenvectors of the normalized Laplacian defined as $L = I_N(1 + \delta) - S$ with $\delta = 0.001$ and $K = 10\%N$ where $N$ is the number of nodes in the original graph. We apply our adapted version of Loukas coarsening algorithm with $n_e = 5\%N$ for SGC Cora, SGC Citeseer and GCN citeseer and $n_e \to \infty$ for GCN Cora (variation edges as defined by Loukas). For SGC cora and SGC Citeseer we make 6 propagations as preprocessing for the features. For GCN Cora and Citeseer we use 2 convolationnal layer with a hidden dimension of 16. For all experiments we use an Adam Optimizer wit a learning rate of 0.05 and a weight decay of 0.01.

### B.6  Hyper-parameters for Table 3

For the experiment on Reddit Dataset, we preserve $K$ eigenvectors of the normalized Laplacian defined as $L = I_N(1 + \delta) - S$ with $\delta = 0.001$ and $K = 400$ eigenvectors to be computationally efficient ($10\%N$ being too big). We apply our adapted version of Loukas coarsening algorithm with

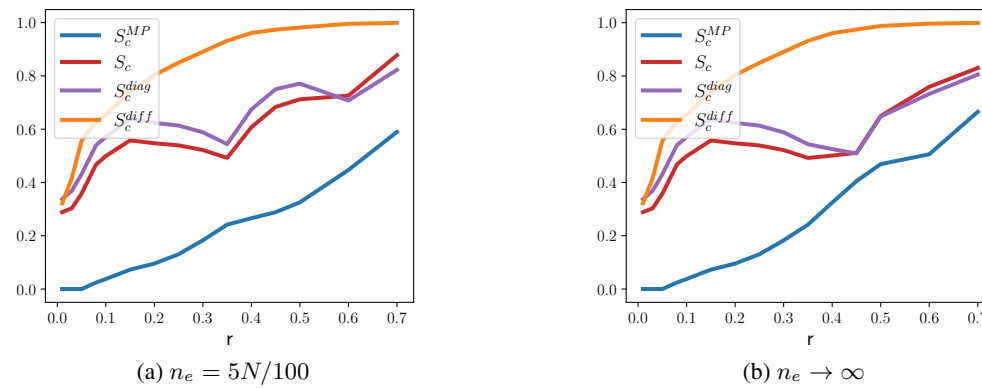

(a) $n_e = 5N/100$  (b) $n_e \to \infty$

Figure 6: Uniform coarsening for Normalized Laplacian

$n_e = 10\%N$ for SGC Reddit and GCN Reddit. We computed 6 propagations for Reddit SGC and 2 for Reddit GCN. We keep the same hidden dimension as for Cora and Citeseer. For the reddit experiments, we use an Adam Optimizer wit a learning rate of $0.1$ and a weight decay of $0.0$.

