# OpenReview forum: "Graph Coarsening with Message-Passing Guarantees"
_NeurIPS.cc/2024/Conference — NeurIPS 2024 poster_

### Official Review · Reviewer_7C68 · 2024-07-02

**Soundness:** 3
**Presentation:** 2
**Contribution:** 4
**Rating:** 6
**Confidence:** 2

**Summary:**

This paper studies the theoretical guarantees of graph coarsening for GNNs. The authors propose a new and directed message-passing operation specific to coarsened graphs, which makes many theoretical results possible.

**Strengths:**

S1: The results are very useful for the field of graph coarsening. Theorem 2 shows the gap of loss between the coarsened graph and the original graph, which is applicable for many tasks.

S2: The new message-passing looks natural and novel to me.

S3: The literature review is comprehensive.

**Weaknesses:**

W1: The overall structure is somewhat loose.

W2: It would be nice to see more intuitive explanation of $S^{MP}_c$, e.g., when $S$ is symmetrically normalized adjacency matrix from GCN.

W3: The experiments are conducted only on small datasets, which I understand since the authors put more effort on the theoretical analysis.

**Questions:**

Q1: Can the authors give some intuitive explanation on $\overline{C}_{\Pi}$, e.g., what kind of coarsening method will make this term smaller?

Q2: It would be nice to see more experiments on larger datasets, e.g., ogbn-arxiv.

Typo: Line 212, $\hat{A}=A+I$.

---

> ### Author Rebuttal · Authors · 2024-08-06
>
> Thank you for your review and feedback. We will fix the typos and clarify the description of the proposed propagation matrix.
>
> **Q1)** *Can the authors give some intuitive explanation on $\overline{C}_{\Pi}$, e.g., what kind of coarsening method will make this term smaller?*
>
> From a theoretical point of view, in general we only have a harsh upper bound: $$C_{\Pi} \leq \lVert S \rVert_{{L}} \sqrt{\frac{\lambda_{max}}{\lambda_{min}}} $$   as $\Pi$ is an orthogonal projector and where $\lambda$ are the eigenvalues of the laplacian ${L}$.
>
> For a propagation matrix $S = \alpha i_n + \beta {L}$ and normalized as it is the case in our experiments $ \lVert S \rVert_{{L}}  \leq 1$. Thus $C_{\Pi}  \leq \sqrt{\frac{\lambda_{max}}{\lambda_{min}}} $.
>
> The same goes with $\overline{C_{\Pi}}  = \lVert \Pi S \Pi \rVert_{{L}}  \leq \lVert \Pi \rVert_{{L}} C_{\Pi} \leq \frac{\lambda_{max}}{\lambda_{min}} $
>
>
> Experimentally we can however remark that we generally observe a factor 10 between the upper bound and the actual value of the constants. Moreover, we obtain lower values for uniform coarsenings. Theoretical interpretation of uniform coarsenings are a promising avenue to use in a coarsening procedure.
>
>
> **Q 2)** *It would be nice to see more experiments on larger datasets, e.g., ogbn-arxiv.*
>
> We have conducted experiments on a larger dataset, namely the Reddit one (see global comment A). As said in the global comment, the Reddit Dataset has 1.5 times more nodes than ogbn-arxiv. For the final version of this paper, experiments on ogbn-arxiv, conducted if time allows, would be a valuable addition.

---

> > ### Comment · Reviewer_7C68 · 2024-08-08
> >
> > Thanks for the response. All my concerns are addressed.

---

### Official Review · Reviewer_eigc · 2024-07-04

**Soundness:** 3
**Presentation:** 3
**Contribution:** 2
**Rating:** 5
**Confidence:** 1

**Summary:**

This paper proposes a new message-passing matrix for a graph coarsening algorithm. The goal is to have some message-passing guarantees for the new message-passing matrix, which is not the case with the previous message-passing matrices based on this coarsening. They provide theoretical proofs for linear variants of GNNs (SGC). They examine their theoretical guarantees on synthetic datasets and do experiments on two real-world graphs for comparing different selections of the message-passing matrices. Their approach in most of the experiments varying in the coarsening ratio, works better than alternative approaches.

**Strengths:**

Graph coarsening can be indeed very beneficial if it can be done efficiently. Pooling approaches in some domains such as computer vision have been very helpful, but they are not as prominent in the learning on graphs community. Having well-studied approaches for this end can be of great importance because of the memory limitations of the GNNs. Also, their work has a deep root in the theory and provides theoretical guarantees for their approach.

**Weaknesses:**

While the theoretical analysis is interesting it is limited to the linear networks. Extending this analysis to more complex GNNs might not be an easy task, however, this does not mean that they could not try their approach for other types of GNNs and see if it works in practice or not. Maybe they could try more common message-passing architecture such as Graph Attention Networks (GAT) or Graph Isomorphism Networks too. Also, the datasets used are fairly old and outdated in the current state of learning on graph works. I would suggest trying some recent datasets, maybe varying from homophilous datasets to heterophyllous ones.

**Questions:**

1. The theory is based on signal processing on graphs that work on a single value for each node. Most datasets have a vector of initial features for each node, how this can be addressed in the theory?

2. How much does the coarsening algorithm help with the memory? GNNs usually scale by the number of nodes + the number of edges, the coarsening ratio r talks about how much you can reduce the number of nodes, but edges seem to be a more important factor in the memory and time complexity. Is there any theoretical or experimental analysis on the edges or memory in general?

**Limitations:**

The theoretical analysis is limited to the linear networks and the experiments are limited to two small datasets.

---

> ### Author Rebuttal · Authors · 2024-08-06
>
> Thank you for your review and feedback.
>
> **Datasets** See global comment (A): we have performed additional experiments on the Reddit Dataset which is significantly bigger than Cora and Citeseer. Concerning heterophilous datasets, we note that spectral-based coarsening itself is probably very inefficient, as it basically aims at preserving the low-frequencies. Dedicated, new coarsening methods is an interesting path for future work. We will add a remark in the paper.
>
> **Other models** Because of attention coefficients, GAT do not rely on a propagation scheme that can be expressed as the multiplication of the node representation $H$ by a fixed propagation matrix $S$. Thus, we can't adapt our method to compute a new propagation matrix for the coarsened graph with this model. However, message-passing on coarsened graphs with attention coefficients and/or edge features is an important path for future work. We will add a remark.
>
>
> **Q 1)** *The theory is based on signal processing on graphs that work on a single value for each node. Most datasets have a vector of initial features for each node, how this can be addressed in the theory?*
>
> See global comment (B).
>
> **Q 2)** *How much does the coarsening algorithm help with the memory? GNNs usually scale by the number of nodes + the number of edges, the coarsening ratio r talks about how much you can reduce the number of nodes, but edges seem to be a more important factor in the memory and time complexity. Is there any theoretical or experimental analysis on the edges or memory in general?*
>
> The memory used and number of edges in the coarsened graph depend on the coarsening algorithm itself: in general, two super nodes are connected if at least two nodes they represent in the original graph are connected. For the Loukas algorithm that we used, Loukas wrote that "The sparsification step was not included in the numerical experiments since it often resulted in increased errors"  which prevent us to perform an additional sparsification step. A new coarsening algorithm that better controls the sparsity of the resulting graphs while balancing with good RSA constant is an interesting path for future work, but out of scope of the present paper.
>
> The number of edges for Reddit after coarsening and cora citeseer can be found in the table 1 (see pdf attached with tables).

---

> ### Comment · Reviewer_eigc · 2024-08-09
>
> I thank the authors for their rebuttal and new experiments.
>
> In general, I think the practical applicability of this work is limited at present. The computational cost seems to be expensive and this class of graph coarsening idea seems to be mostly applicable to homogeneous datasets (usually less challenging datasets), and as nodes inside a supernode can only be assigned to the same class, the performance would be poor on more heterogeneous datasets.
>
> However, I also think that we need more theory to understand the coarsening and pooling algorithms. I am not an expert in this area and cannot evaluate the significance of the theoretical results provided in this work or their applicability to other works or for future theoretical analysis. Relying on Reviewer 7C68's review, I would like to increase my score to 5; however, I want to decrease my confidence level in my assessment to 1, since I am making a decision in an area where my knowledge is very limited.

---

> ### Author Response · Authors · 2024-08-13
>
> Thank you for your answer. Indeed, your are correct in pointing out that the *coarsening* process in itself is still an active area of research and that classical spectral-based coarsening must be improved for certain datasets. Our work, however, studies message-passing on coarsened graphs, which is downstream from the coarsening process. But we hope that it might serve as pointers to improve the coarsening itself in future work.

---

### Official Review · Reviewer_foGd · 2024-07-10

**Soundness:** 3
**Presentation:** 2
**Contribution:** 3
**Rating:** 7
**Confidence:** 4

**Summary:**

This work presents a novel computation method for the message-passing matrix on coarsened graphs. This method does not require recalculating degree matrices and other information on the coarsened graph and has comprehensive theoretical guarantees. Overall, it addresses a significant problem in graph coarsening field.

**Strengths:**

1. The theoretical analysis is sufficient and reasonable.
2. The proposed method is very simple.
3. The model performs very well on Cora and Citeseer.

**Weaknesses:**

The experimental section is insufficient. Testing only on Cora and Citeseer is not enough to demonstrate the effectiveness of the method. More GNN models should also be tried. I believe this work is a valuable contribution to the field of graph coarsening, and if the authors further increase the experiments, I will raise my score.

**Questions:**

How effective is this work on datasets such as Arxiv and Products?

**Limitations:**

The authors adequately addressed the limitation and potential negative societal impact  of their work.

---

> ### Author Rebuttal · Authors · 2024-08-06
>
> Thank you for your review and feedback.
>
> **Q 1)** *How effective is this work on datasets such as Arxiv and Products*
>
> See global comment (A): improving our spectral corsening algorithm, we have conducted experiments on a larger graph, Reddit. This graph has 1.5 times more nodes than ogb-arxiv. For the final version of this paper, experiments on ogbn-arxiv,  conducted if time allows, would be a valuable addition.

---

> > ### Comment · Reviewer_foGd · 2024-08-10
> >
> > Thanks for the response. I believe that the volume of experiments in this paper still has not reached my expectations. However, considering that the paper addresses a very important issue in graph coarsening, I have raised the score to 7 and encourage the authors to continue adding more experiments.

---

> > > ### Author Response · Authors · 2024-08-13
> > >
> > > Thank you for your answer. We are confident at this point that it will be possible for us to extend the experimental section with other large datasets beyond Reddit.

---

### Official Review · Reviewer_GXdc · 2024-07-11

**Soundness:** 2
**Presentation:** 3
**Contribution:** 2
**Rating:** 6
**Confidence:** 5

**Summary:**

The authors describe an alternative way to obtain the connectivity matrix of a coarsened graphs and provide some bounds on operations performed on such a matrix.

**Strengths:**

Theoretical work on how to optimally compute the connectivity matrix of a coarsened graph is an interesting and potentially useful research direction.

**Weaknesses:**

- The main contribution is very small, as it simply consists in replacing the coarsened matrix QSQ^T, commonly used in graph pooling, with QSQ^+. This seems more of a detail in practice and it seems too much to have a whole paper on it. I seriously doubt it would make a significant difference in practice and the limited experimental evaluation (more on this later) does not help to address my concern.
- I don't see the usefulness of Theorems 1 and 2, which is the second contribution of the paper. They provide bounds which I don't find useful, as they do not compare against other existing bounds and they are not computed for other coarsening schemes, such as the more common QSQ^T. For example, it would be useful to see that the proposed coarsened matrix yields narrower bounds than the latter.
- I believe that there are too many simplifications and assumptions for the theoretical results to be relevant in practice. For example, the RSA constant is defined only for 1-D node features, which is something not commonly found in many graph data processed by graph neural networks. Similarly, it seems that Theorem 2 relies on the assumption that each column of the node features X belongs to R, which seems too strong and unrealistic as assumption. Finally, the whole paper assumes GNNs without nonlinearities. I believe that a GNN without nonlinearity is not a GNN and a paper completely centered around the analysis of such models should rather be published in a (graph) signal processing or linear algebra venue, not a machine learning one.
- The experimental evaluation is too limited and not convincing. First of all, it only considers relatively small graphs, as the coarsening algorithm used does not scale well. This defies the whole premise of using coarsened graphs to handle large graphs that cannot be processed due to high computational complexity.
- Only one coarsening algorithm is considered to obtain Q, while there is a large pletora of existing graph pooling algorithms that can be used to compute Q. To convince about the effectiveness of the proposed method, the author(s) should show that it works with different coarsening schemes. Remarkably, the coarsening algorithm used in the experimental evaluation does not even account for node features. This, again, set the work apart from the GNN and machine learning community.
- Besides the synthetic data, the only two datasets considered are Cora and Citeseer. These datasets are rather similar (both of them are citation networks) and they have very homophilic node features\labels, which might biases the experimental evaluation. In addition, the experiment considers only the largest connected component of these networks, which further simplify the task on datasets that are already simplicistic. The need for such an unusual experimental setting casts further doubts on the effectiveness of the proposed method.
- The authors largely overlook relevant related work on graph pooling. See for example the CONNECT operation from the paper entitled "Understanding Pooling in Graph Neural networks".
- (minor) Ker(L) is not defined I think.

**Questions:**

I do not have questions besides the concerns above.

**Limitations:**

See what I wrote in weaknesses.

---

> ### Author Rebuttal · Authors · 2024-08-06
>
> Thank you for your review and feedback. We have addressed your comments individually below. Before answering the questions, we would like to precise that Graph Pooling and Graph coarsening are two methods that are linked, but with different purposes. Graph Pooling is generally incorporated into the GNN to upgrade the classification results by mimicking the "pooling" in CNNs. It is often supervised, and diferentiable, such as DiffPool, but in turns has generally no guarantees, being the result of a non-convex optimization problem. On the other hand, Graph coarsening is a preprocessing used to gain memory. It is often non-supervised, with the type of guarantees such as the RSA constant introduced by Loukas.
>
> **Q 1)** *The main contribution is very small, as it simply consists in replacing the coarsened matrix $QSQ^T$, commonly used in graph pooling, with $QSQ^+$.  [...]*
>
> We agree that this new propagation matrix is indeed "deceptively" simple, even if it is quite original in the sense that it is not a Laplacian or any other form of graph representation matrix, as it may not even be symmetric. In effect, we show that our proposal is the "right" normalization to directly translate spectral guarantees (which are classical in graph coarsening and the objective of many coarsening algorithms), to MP guarantees. Other matrices simply do not have this mathematical property. Note that when $Q$ is orthogonal, the two coincide, but classical "uniform" coarsenings are not orthogonal.
>
> Also note that we compared in the experiments our matrix with $S\_c^{diff} =QSQ^{T}$ inspired by the pooling litterature, and in our experiments, $S_c^{diff}$ give accuracy results which are far less competitive.
>
>
> **Q 2)** *I don't see the usefulness of Theorems 1 and 2, [...] They provide bounds which I don't find useful, as they do not compare against other existing bounds and they are not computed for other coarsening schemes, such as the more common $QSQ^T$. [...]*
>
> The crucial part of our theoretical bounds is to the dependency on the RSA-constant $\epsilon_{Q, {L}, \mathcal{R}}$, which tend to be small, as it is explicitely minimized by coarsening algorithms. Unfortunately, other propagation matrices (including $QSQ^T$) simply do *not* yield any mathematical guarantee in this spectral-based framework, hence the impossibility to ``compare'' with existing bounds. We will clarify this in the final version of the paper.
>
> **Q 3)** *I believe that there are too many simplifications[...]. For example, the RSA constant is defined only for 1-D node features[...]. Similarly, it seems that Theorem 2 relies on the assumption that each column of the node features X belongs to R [...] Finally, the whole paper assumes GNNs without nonlinearities*
>
> See global comment (B) for multidimensional node features. Thm 2 relies on the assumption that each column of the nodes features $X_{:,i} \in \mathcal{R}$, that is, are close to low-frequency. This assumption seems reasonable for homophilic datasets (Cora, Citeseer) and large preserved space.
>
> We agree that for now the assumption on non-linearities is strong. However SGC is indeed used in many theoretical works to analyse the inner workings of GNNs (see eg [1,2] and references therein), and we still believe that it opens a path for interesting future works on the interaction between low-frequencies and non-linearities.
>
> **Q 4)** *The experimental evaluation [...] only considers relatively small graphs, as the coarsening algorithm used does not scale well. This defies the whole premise of using coarsened graphs to handle large graphs [...].*
>
> See global comment (A).
>
> **Q 5)** *Only one coarsening algorithm is considered to obtain Q, while there is a large pletora of existing graph pooling algorithms that can be used to compute Q. [...] Remarkably, the coarsening algorithm used in the experimental evaluation does not even account for node features.[...]*
>
> The starting point of our work are indeed coarsening algorithm that present spectral-based theoretical bounds (see header comment on graph pooling vs graph coarsening). To our knowledge, the Loukas coarsening algorithm is one of the most classical algorithm with such theoretical spectral guarantees. We do agree that future works focusing on the nodes features and coarsening algorithms incorporating both spectral guarantees and nodes features are a promising avenue, but this paper did not focus on the coarsening algorithm itself, but rather how to translate classical spectral guarantees to GNNs.
>
> **Q 6)** *Besides the synthetic data, the only two datasets considered are Cora and Citeseer. These datasets are rather similar (both of them are citation networks) and they have very homophilic node features\labels, which might biases the experimental evaluation. In addition, the experiment considers only the largest connected component of these networks [...]*
>
>  We considered the main connected component of Cora and Citeseer as Loukas coarsening algorithm was designed for fully connected graph. It is true that spectral coarsening algorithms might be less efficient on heterophilic datasets, since the features are less likely to live in the low frequency of the graph. The design of new coarsening algorithms for these graphs is an important path for future work. We conducted additional experiments on the Reddit Dataset with good results, see global comment (A)
>
> **7)** *See for example the CONNECT operation from the paper entitled "Understanding Pooling in Graph Neural networks"*
>
> Thank you for the suggestion, we will add this reference to the final version and discuss it. We will also clarify the relation between pooling and coarsening (see top).
>
> [1] Zhu et al. *Graph Neural Networks with Heterophily*. AAAI.
>
> [2] Keriven. *Not too little, not too much: a theoretical analysis of graph (over)smoothing*. NeurIPS.

---

> > ### Comment · Reviewer_GXdc · 2024-08-10
> >
> > Thank you for the detailed answers.
> >
> > I do not completely agree with the distinction proposed by the authors between graph pooling and graph coarsening. Graph pooling can also be a pre-processing step that reduces the size of the graph and, thus, the memory consumption. See for example Graclus [1], originally introduced by [2] as a pooling scheme, and other non-trainable pooling operators described in [3]. In addition, there are some recent work that showed some guarantees of pooling operators in terms of their capability of keeping two non-homomorphic graphs distinguishable after pooling [4]. I suggest the authors to clarify this connection... or to find a stronger argument for why pooling and coarsening should be different things.
> >
> > I still believe that the practical contribution seems a rather small detail that would arguably make a small difference using $QSQ^+$ rather than $QSQ^T$ in most practical settings. At least, that's the experience I had myself when I replaced $S^T$ with a pseudo-inverse on some problems I am currently working with. Nevertheless, I see that the value of this work is to be a starting point for a theoretical study on pooling/coarsening in GNNs that will hopefully be developed further in the future.
> >
> > Even if most of my concerns still remain after the rebuttal, I do appreciate the effort of the authors in answering in detail to every author and adding additional experiments. Therefore, I'll raise my scores conditional on the fact that the authors will modify the paper as asked.
> >
> > [1] I. S. Dhillon, Y. Guan, and B. Kulis. Weighted graph cuts without eigenvectors a multilevel
> > approach. IEEE transactions on pattern analysis and machine intelligence, 29(11):1944–1957,
> > 2007.
> >
> > [2] M. Defferrard, X. Bresson, and P. Vandergheynst. Convolutional neural networks on graphs
> > with fast localized spectral filtering. In Advances in Neural Information Processing Systems,
> > pages 3844–3852, 2016.
> >
> > [3] Grattarola, D., Zambon, D., Bianchi, F. M., & Alippi, C. (2022). Understanding pooling in graph neural networks. IEEE transactions on neural networks and learning systems, 35(2), 2708-2718.
> >
> > [4] Bianchi, F. M., & Lachi, V. (2024). The expressive power of pooling in graph neural networks. Advances in neural information processing systems, 36.

---

> ### Author Response · Authors · 2024-08-12
>
> Thank you for your careful review and for raising your score.
>
> A few comments on the points you mention.
>
> **On pooling**: we agree that the vocabulary in the community might overlap a bit at this point. We'll try to clarify as much as we can our meaning in the final version (that is, our focus on spectral-based unsupervised coarsening), and add the references you mention. Thank you for providing them.
>
> **On propagation matrix**: it is true that the ``more orthogonal'' $Q$ is, the less the difference between $Q^+$ and $Q^T$. More generally, the difference between the two is more pronounced when supernodes are of very different sizes, which may happen for highly irregular graphs (eg, for uniform coarsenings, when all the supernodes are exactly of the same size, there is only a multiplicative constant between $Q^+$ and $Q^T$). We will explain this better in the final version, as well as outline the datasets where this happens more frequently.

---

> > ### Comment · Reviewer_GXdc · 2024-08-13
> >
> > The point you made in your answer about the propagation matrix, i.e., that $Q^{+}$ and $Q^\top$ become more similar as $Q$ induces a balanced partition makes sense but it is something I missed when reviewing the paper. Indeed, I strongly encourage the authors to stress this point.
> >
> > There is a class of graph pooling methods that encourage the size of the supernodes to be balanced (see for example the dense pooling methods from this recent survey paper [1]). In this case, it would make less sense to use $Q^{+}$. Again, this seems an important point worth commenting on.
> >
> > [1] Wang, Pengyun, et al. "A Comprehensive Graph Pooling Benchmark: Effectiveness, Robustness and Generalizability." arXiv preprint arXiv:2406.09031 (2024).

---

> > > ### Author Response · Authors · 2024-08-13
> > >
> > > Thank you for the pointers and reference, that is indeed an important point that we will emphasize in the final version.

---

> > > > ### Comment · Reviewer_GXdc · 2024-08-13
> > > >
> > > > I further improved my score, trusting the authors to implement all the modifications they promised to me and the other reviewers.

---

### Official Review · Reviewer_5Vcn · 2024-07-13

**Soundness:** 3
**Presentation:** 3
**Contribution:** 3
**Rating:** 5
**Confidence:** 3

**Summary:**

This paper proposes a novel message-passing guarantee for graph coarsening and a new message-passing operation with the message-passing guarantee. Experiments demonstrate that the prediction performance of the proposed method outperforms some baselines.

**Strengths:**

1. The proposed message-passing guarantee is novel.
2. The authors provide the theoretical analysis

**Weaknesses:**

1. How to select the hyperparameters in experiments (e.g. the number of the SGC layers)? The selected coarsening ratio is significantly larger than existing works.
2. Do linear GNNs [3] satisfy Assumption 4?
3. I am not sure whether the analysis under the linearity assumption is enough. Assume the processed features of SGC is $H=A^TX \in \mathbb{R}^{n \times r}$, where the feature dimension $r$ is significantly smaller than the number of nodes $n$. By noticing that the rank of $H$ is lower than $r$, we can compress the node features into sizes $(r,r)$ without errors. So, what is the motivation for graph coarsening under the linearity assumption?
4. The authors may want to compare the spectral guarantee and the proposed message-passing guarantee in details. Moreover, I suggest summarizing these theoretical properties of existing methods and the proposed method.
5. The formulation of message passing is different from [5]. The message passing framework considers the graphs with edge features while Equation (1) does not consider them. Therefore, the concept of message passing guarantees may mislead readers. In my opinion, the proposed concept in this paper is close to convolution matching [6].
6. How to effectively compute $Q^+$ in practice? The complexity analysis is missing.


[1] Featured Graph Coarsening with Similarity Guarantees.

[2] Graph Distillation with Eigenbasis Matching.

[3] How Powerful are Spectral Graph Neural Networks?

[4] Graph Reduction with Spectral and Cut Guarantees.

[5] eural Message Passing for Quantum Chemistry.

[6] Graph Coarsening via Convolution Matching for Scalable Graph Neural Network Training

**Questions:**

See Weaknesses.

**Limitations:**

The authors have adequately addressed the limitations.

---

> ### Author Rebuttal · Authors · 2024-08-06
>
> Thank you for your review and feedback. We address each comment below.
>
> **Q 1)** *How to select the hyperparameters in experiments (e.g. the number of the SGC layers)? The selected coarsening ratio is significantly larger than existing works*
>
>
> We chose classical values for the training of SGC models as the paper don't focus on that point but on the theoretical guarantees. We will add more range of parameters, including coarsening ratios, in the appendices of the final version.
>
> **Q 2)** *Do linear GNNs [3] satisfy Assumption 4?*
>
> A linear GNN is formulated in [3] as $Z = g(\hat{L})XW$ where $Z \in \mathbb{R}^{N \times d}$ is the prediction matrix, $g$ is a learnable real-valued polynomial and $W \in \mathbb{R}^{N \times d}$ a learnable weight matrix. The polynom of $L$ is a message passing with a number of layers equal to the degrees; as it is linear with $\sigma = id$, hence the linear GNNs satisfies assumption 4. We will add the reference.
>
>
> **Q 3)** *I am not sure whether the analysis under the linearity assumption is enough. Assume the processed features of SGC is $H = A^KX \in \mathbb{R}^{n\times d}$, where the feature dimension is significantly smaller than the number of nodes . By noticing that the rank of is lower than $n$ , we can compress the node features into sizes without errors. So, what is the motivation for graph coarsening under the linearity assumption?*
>
> Thank you for this very interesting remark. Compressing the propagated features into sizes of the nodes features ranks would indeed result in a very different but efficient compressing method, even if it somewhat blurs the link with vanilla semi-supervised learning as the loss couldn't be computed directly. We elected to keep a direct link with message-passing and GNNs in this work, but consider your suggestion as a promising avenue for a future work.
>
>
> **Q 4)** *The authors may want to compare the spectral guarantee and the proposed message-passing guarantee in details. Moreover, I suggest summarizing these theoretical properties of existing methods and the proposed method.*
>
>
> With our new propagation matrix  $ S^\textup{MP}\_{c}$ on the coarsened graph (contrarly to other choices), spectral guarantees *lead* to message passing guarantees through the RSA-constant : $\epsilon\_{Q,{L}, \mathcal{R}}$. Hence they are one and the same. To our knowledge, no other method has guarantees of this type, hence our difficulty to ``compare'' the theoretical bounds.
>
> **Q 5)** *The formulation of message passing is different from [5]. The message passing framework considers the graphs with edge features while Equation (1) does not consider them. Therefore, the concept of message passing guarantees may mislead readers. In my opinion, the proposed concept in this paper is close to convolution matching [6].*
>
> Thank you for this remark. We agree that incorporating edge features in coarsened graphs in an important path for future work (eg to handle GAT), however GNNs without edge features remain classical in a lot of fundamental models such as GCNconv or GIN, where the auhors effectively define the "propagation matrix" to describe the message-passing process.
>
> **Q 6)** *How to effectively compute $Q^+$ in practice*
>
> We use Loukas coarsening algorithm, where the matrix $Q$ is "well defined", i.e each original node is mapped to an unique "super node" in the coarsened graph. That means that $Q$ has an unique non zero entries by column.
>
> For such matrix $Q \in \mathbb{R}^{N,n}$, Loukas proposed an "easy inversion" property (Proposition 6) $ Q^{+} = Q^TD^{-2}$ with $D(r,r) = \lVert Q(r,:) \rVert\_2 $.
>
> As the matrix Q has $N$ non-zero entries and as it is needed to compute the sum of the rows of Q, the complexity to compute the pseudo inverse is linear with the number of nodes in the original graph $\mathcal{O}(N)$
>
> [3] How Powerful are Spectral Graph Neural Networks?
>
> [5] Neural Message Passing for Quantum Chemistry.
>
> [6] Graph Coarsening via Convolution Matching for Scalable Graph Neural Network Training

---

> > ### Comment · Reviewer_5Vcn · 2024-08-10
> >
> > Thanks for your rebuttal. The rebuttal has addressed Concerns 2 and 6. Unfortunately, my concerns about Concerns 1,3,4,5 remain unaddressed. The suggestions and questions are as follows.
> >
> > 1. **Why is the selected coarsening ratio significantly larger than existing works?** How to select the coarsening ratio in experiments?
> > 2. **What is the motivation** for graph coarsening under the linearity assumption?
> > 3. The answer for Concern 4 is confusing. If spectral guarantees lead to message-passing guarantees, then what is the contribution of this paper? A method with spectral guarantees is enough in practice, as the method also ensures message-passing guarantees.
> > 4. The concept of message-passing guarantees is still confusing. Given your answer, convolution guarantees may be more accurate than message-passing guarantees, as the convolution operation usually does not consider the edge features.

---

> ### Author Response · Authors · 2024-08-12
>
> Thank you for your answer. We attempt to answer your remaining questions below.
>
> 1. *Why is the selected coarsening ratio significantly larger than existing works? How to select the coarsening ratio in experiments?*
>
> We select three coarsening ratios that are used in Loukas work [1] to illustrate our theoretical results. Following your suggestion, the final version of the paper will include more coarsening ratios (please note that in the Dickens [4] work mentioned earlier in your questions, the coarsening ratio is defined as $1-r$ compared to Loukas and ours). In practice, selecting the ratio really depends on the use-case, whether the aim of the user is to save storage memory, train a GNN, and so on. For instance, compared to smaller datasets, we had to select a very high coarsening ratio for Reddit in order to train a GNN for it on a laptop. Of course, the higher the ratio, the ``worse'' the results compared to the original graph.
>
> 2. *What is the motivation for graph coarsening under the linearity assumption?*
>
> We agree that for now the assumption on non-linearities is strong. However SGC is indeed used in many theoretical works to analyse the inner workings of GNNs (see eg [2,3] and references therein), and we still believe that it opens a path for interesting future works on the interaction between low-frequencies and non-linearities, in order to treat more general GNNs on coarsened graphs.
>
> 3. *The answer for Concern 4 is confusing. If spectral guarantees lead to message-passing guarantees, then what is the contribution of this paper? A method with spectral guarantees is enough in practice, as the method also ensures message-passing guarantees.*
>
> Spectral guarantees and message-passing are of different nature, and concern different objects. Spectral guarantees are inherent to a graph coarsening, and refer to the fact that low-frequencies of the graphs are preserved by coarsening (aka a low $\epsilon$ constant). Most algorithms, such as Loukas' that we employ in the experiments, aim at producing such spectral guarantees. Message-passing guarantees concerns the choice of a *propagation matrix*. Our work consist in showing that, even assuming that the coarsening exhibit spectral guarantees, message-passing guarantees are *not* automatic, and generally not satisfied for naive choices of propagation matrices. We then propose a new propagation matrix that yields such message-passing guarantees, when the coarsening has spectral guarantees (that is, we bound the message-passing error of this propagation matrix by $\epsilon$, hence the fact that spectral guarantees ``lead to'' message-passing guarantees *for this new propagation matrix only*). We will clarify this in the final version.
>
> 4. *The concept of message-passing guarantees is still confusing. Given your answer, convolution guarantees may be more accurate than message-passing guarantees, as the convolution operation usually does not consider the edge features.*
>
> Thank you for your suggestion. We will not change the title at this point, but will make this point of vocabulary clear in the final version.
>
> [1] Graph Reduction with Spectral and Cut Guarantees, Andreas Loukas, JMLR 2019
>
> [2] Zhu et al. Graph Neural Networks with Heterophily. AAAI.
>
> [3] Keriven. Not too little, not too much: a theoretical analysis of graph (over)smoothing. NeurIPS.
>
> [4] Dickens et al. Graph Coarsening via Convolution Matching for Scalable Graph Neural Network Training

---

> > ### Comment · Reviewer_5Vcn · 2024-08-13
> >
> > Thanks for the detailed response and most of my concerns have been addressed. Therefore, I raise my score to support the acceptance of this paper.

---

### Author Rebuttal · Authors · 2024-08-06

We thank all the reviewers for their reviews and questions. In this global comment, we address two questions that were mentioned in multiple reviews. Namely, we introduce new experiments on a larger dataset (Reddit) and comment on the multidimensionality of node features.

**A)** ***Larger Dataset***


As mentioned in the paper, Loukas' coarsening algorithm is one of the only one that provides RSA guarantees, but in turns it is quite costly to run. Improving the computational cost of spectral-based coarsening algorthms is a major path for future work, but out of scope of this paper. Hence we evaluated our coarsened propagation matrix proposition on small-ish datasets for the first version of our paper. During the reviewing process, we upgraded our code to deal with larger graphs and have conducted experiments on the Reddit Dataset, which is 1.5 bigger than ogb-arxiv dataset and 100 times bigger than Cora or Citeseer.

We performed two coarsening ratio on the Reddit Dataset ( available in torch geometric) $ r = 90\%$ and $ r = 99\%$, their number of nodes, and edges can be found in the table 1 (see additional pdf with tables). For a final version paper, experiments will be conducted on ogbn-arxiv if time allows.

The coarsening is performed with the adaptation of Loukas coarsening variation edges and preserving the first 400 eigenvectors. As a reference point for such heavy coarsening ratios, we add the "max acc possible", which is inherent to the coarsening: it corresponds to the optimal prediction over the super-nodes of the coarsened graph (all the nodes coarsened in a super nodes has the same prediction, optimally the majority label of this cluster). For the node classification task, the corresponding learning rate and weight decay are 0.1 and 0.0.

The node prediction results on the Reddit Dataset are reported in table 2. Our propagation matrix  ${S^\textup{MP}_c}$ achieves very good results with the SGC model, very close to the maximum accuracy possible on the given coarsening. Our propagation matrix is still competitive with the GCNconv model and achieved better results on the biggest coarsening ratio.

The Message-Passing error for different coarsening propagation matrices is reported in table 3. Our propagation matrix for coarsened graphs achieved a better Message-Passing error, close to the RSA-constant computed in the coarsen graph. It is consistent with the fact the Message-Passing error is bounded by theorem 1 with our propagation matrix, and we thus expect lower values.

Thus, these additional experiments show better the effectiveness of our method on large graph for which coarsening as a preprocessing is crucial to save memory.


**B)** ***Multiple dimension node features***

The spectral guarantees and the corresponding RSA-constant $\epsilon_{Q , {L}, \mathcal{R}}$ depend of a *whole vector space* $\mathcal{R} \subset \mathbb{R}^{N}$. It therefore applies to multidimensional features by treating each coordinates independently, as seen in Theorem 2. We will clarify this in the final version.

---

### Decision · Program_Chairs · 2024-09-25

**Decision:**

Accept (poster)

**Comment:**

In the context of GNNs with pooling, the authors propose computing the coarse graph from the full graph $A$ using the pseudoinverse of the coarsening matrix instead of its transpose in the formula $Q^TAQ$. This construction comes with theoretical guarantees: training a GNN on the coarsened graph is provably close to training it on the original graph. While the practical advantage of replacing the coarse graph $Q^TAQ$ by $Q^+AQ$ may be marginal, the theoretical analysis done by the authors is novel and brings new insights into the graph coarsening/pooling community.

The authors are asked to incorporate the reviewer comments for the camera ready version.